# SEMANTIC REGEXES: AUTO-INTERPRETING LLM FEATURES WITH A STRUCTURED LANGUAGE

**Angie Boggust**
MIT CSAIL
Cambridge, MA, USA
aboggust@csail.mit.edu

**Donghao Ren**
Apple
Seattle, WA, USA
donghao@apple.com

**Yannick Assogba**
Apple
Cambridge, MA, USA
yassogba@apple.com

**Dominik Moritz**
Apple
Pittsburgh, PA, USA
domoritz@apple.com

**Arvind Satyanarayan**
MIT CSAIL
Cambridge, MA, USA
arvindsatya@mit.edu

**Fred Hohman**
Apple
Seattle, WA, USA
fredhohman@apple.com

## ABSTRACT

Automated interpretability aims to translate large language model (LLM) features into human understandable descriptions. However, natural language feature descriptions can be vague, inconsistent, and require manual relabeling. In response, we introduce *semantic regexes*, structured language descriptions of LLM features. By combining primitives that capture linguistic and semantic patterns with modifiers for contextualization, composition, and quantification, semantic regexes produce precise and expressive feature descriptions. Across quantitative benchmarks and qualitative analyses, semantic regexes match the accuracy of natural language while yielding more concise and consistent feature descriptions. Their inherent structure affords new types of analyses, including quantifying feature complexity across layers, scaling automated interpretability from insights into individual features to model-wide patterns. Finally, in user studies, we find that semantic regexes help people build accurate mental models of LLM features.

## 1 INTRODUCTION

Large language models (LLMs) represent their learned concepts, like the "*Golden Gate Bridge*", as linear directions in latent space called *features* (Bricken et al., 2023). Understanding a model's features helps us anticipate its behavior, assess its alignment, and intervene to ensure safe outcomes (Templeton et al., 2024). Automated interpretability assigns human-readable descriptions to each feature by analyzing patterns in its response to input data (Bills et al., 2023; Lin, 2023; Paulo et al., 2024). With feature descriptions, researchers have identified how models encode domain-relevant concepts, like protein structures (Gujral et al., 2025), and reconstructed circuits of features that correspond to complex behaviors, like medical diagnoses (Lindsey et al., 2025).

Despite the advantages of feature descriptions, natural language is an imprecise interface for describing the computational roles features play in a model's inference. Current automated interpretability methods often yield overly verbose or inconsistent descriptions (Huang et al., 2023), reflecting the difficulty of capturing tightly bounded feature behaviors in free-form text. Moreover, because natural language is prone to ambiguity, it is poorly suited for interpretability tasks that require compositional reasoning, such as studying feature complexity or identifying redundant features. As a result, even recent work identifying feature circuits in LLMs report needing skilled human relabeling to describe a feature's role in the network (Ameisen et al., 2025).

In contrast to natural language, structured languages (e.g., regular languages, programming languages) offer well-defined syntax and semantics (Chomsky, 1956). By combining a set of primitives with compositional rules, structured languages capture precise patterns while maintaining expressivity (Lawson, 2005). Their grammatical structure provides additional affordances, including consistent ways of expressing the same pattern (Knuth, 1965), concise representations of complex

patterns (Lawson, 2005), and mechanisms for representing abstraction (Backus, 1959; Aho & Ullman, 1972). As a result, they can construct complex yet specific expressions, ranging from regular expressions (Lawson, 2005) to software (Python Software Foundation, 2025).

To gain the benefits of structured language in automated interpretability, we introduce *semantic regexes*. The semantic regex language is designed to capture the diverse activation patterns of LLM features, while providing the additional affordances of structured language. Its primitives are grounded in commonly observed feature functions, including exact token matches (`symbols`), syntactic variants of words and phrases (`lexemes`), and broader semantic relationships (`fields`). To enable greater expressivity, we extend these primitives with modifiers for context, composition, and quantification. As a result, semantic regexes can express a range of features, from simple token detectors (e.g., "*on*"→`[:symbol on:]`) to complex linguistic phenomena (e.g., "*the last name of a politician when it follows their title*"→`[:field political title:][:field last name:]`).

Across automated interpretability evaluations, we find that semantic regexes are as accurate as natural language, showing that constraining feature descriptions to the semantic regex language does not reduce expressivity. Beyond accuracy, their structured form offers additional affordances for interpretability, including producing more concise feature descriptions and enforcing description consistency across functionally similar features. Moreover, since semantic regex primitives and modifiers exist across levels of abstraction, they serve as a proxy for feature complexity, allowing us to perform model-wide analysis of feature behavior. Finally, in a user study, we find that semantic regexes help people build accurate mental models of LLM features.

## 2 RELATED WORK

Mechanistic interpretability exposes LLMs' internal concepts to explain their behavior and evaluate their alignment (Olah et al., 2017; 2020; Bricken et al., 2023). LLMs encode concepts along linear directions in latent space, often referred to as features (Elhage et al., 2022; Park et al., 2024). Methods like sparse autoencoders (SAEs) (Huben et al., 2024; Rajamanoharan et al., 2024; Bussmann et al., 2025; Gao et al., 2025) and transcoders (Dunefsky et al., 2024; Ameisen et al., 2025) uncover human-interpretable features, including those that correspond to domain-specific and safety-relevant concepts (Bricken et al., 2023; Gujral et al., 2025; Lindsey et al., 2025).

While features correspond to concepts, *which* concept a feature encodes is not obvious a priori. Early interpretability approached this problem by manually analyzing the data that activate each feature (Olah et al., 2017; 2018; 2020; Carter et al., 2019). To scale this process, automated interpretability prompts a language model to describe each feature based on its activating data (Bau et al., 2017; Hernandez et al., 2022; Bills et al., 2023; Lin, 2023; Shaham et al., 2024; Paulo et al., 2024; Gur-Arieh et al., 2025). Aligned with this goal, our method adopts a similar pipeline for generating feature descriptions, but generates descriptions using a structured language.

Our approach is inspired by recent research showing that LLM features represent structured concepts. Activation analyses show that feature complexity increases with depth (Jin et al., 2025) and models represent concepts across levels of abstraction (Chanin et al., 2024; Boggust et al., 2025; Bussmann et al., 2025; Zaigrajew et al., 2025). In response, researchers have proposed lightweight taxonomies that classify features by function (Ameisen et al., 2025; Lindsey et al., 2025; Gur-Arieh et al., 2025), and called for feature description formalisms (Huang et al., 2023). These findings motivate our use of a structured language, enabling semantic regexes to describe a feature's activation pattern and express its level of abstraction.

## 3 SEMANTIC REGEXES

Semantic regexes use a structured language to describe LLM features. Built around a system of human-interpretable *primitives* (Section 3.1.1) and *modifiers* (Section 3.1.2), semantic regexes capture the low-level syntactic patterns and higher-level semantic concepts that LLM features represent. Unlike natural language, which is flexible but ambiguous, the semantic regex language restricts expressivity to ensure the resulting feature descriptions explicitly convey their meaning. On the other hand, while inspired by regular expressions, the semantic regex language is not a regular language and extends beyond one-to-one character patterns to capture more abstract concepts.

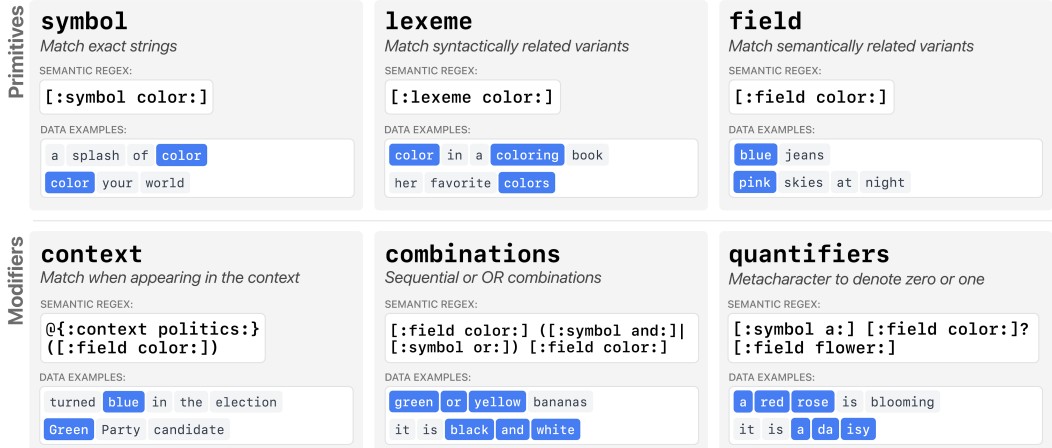

Figure 1: The semantic regex language consists of a set of primitives (top) that can be applied independently or combined with modifiers (bottom) to express diverse feature activation patterns.

## 3.1 THE SEMANTIC REGEX LANGUAGE

The semantic regex language consists of compositional components: *primitives* define the textual units a semantic regex matches, and *modifiers* refine or expand their scope (Figure 1). Together these components form a compact, yet expressive language for specifying LLM feature patterns.

We developed the semantic regex language using a grounded-theory approach (Corbin & Strauss, 1998), deriving its components from empirical analysis of real LLM features. By manually surveying thousands of features across models, layers, and feature sources on Neuronpedia (Lin, 2023), we identified recurring patterns (e.g., context dependent activations). We introduced new primitives or modifiers when they captured a recurring pattern, increased the language's descriptive coverage, and preserved intelligibility of the language. We continued this iterative process until reaching saturation, resulting in a language capable of describing all the features we examined.

### 3.1.1 PRIMITIVES

Primitives are the atomic components of a semantic regex. They specify the type of textual pattern a semantic regex will match, ranging from exact characters to categorical relationships.

**Symbols**  Symbols (`[:symbol X:]`) match exact strings `X`. For instance, `[:symbol color:]` matches the string "*color*", such as in a splash of color. These are the most specific and simplest primitives, and they commonly describe features that activate on specific tokens or phrases.

**Lexemes**  Lexemes (`[:lexeme X:]`) match syntactic variants of `X`. Drawing on linguistics, a lexeme is the abstract form of a word that encompasses all of its surface variants, like changes in tense or plurality. For example, `[:lexeme color:]` matches "*color*", "*colors*", "*coloring*", etc., such as color in a coloring book. Lexemes typically describe features that capture a word's meaning.

**Fields**  Fields (`[:field X:]`) match semantic variants of `X`. Drawing on linguistics, fields refer to words or phrases in a conceptual domain. For instance, `[:field color:]` matches "*red*", "*blue*", etc., as in blue jeans. Fields often apply to features that activate on a conceptual category.

### 3.1.2 MODIFIERS

Modifiers refine and extend primitives with context, composition, and quantification. This increases the expressive power of semantic regexes, allowing them to represent a wider range of features.

**Context** Contexts (`@{:context X:}(semantic regex)`) match a semantic regex in the context X. For instance, `@{:context politics:}([:symbol color:])` only matches `[:symbol color:]` in a political context, matching `Green` `Party` but not `green` `apple`. Contexts help match features that represent domain-dependent concepts.

**Composition** Semantic regexes can compose in *sequence* or *alternation* (`|`). For instance, `[:field color:]([:symbol and:]|[:symbol or:])[:field color:]` matches both `green` `or` `yellow` and `black` `and` `white`. Composing semantic regexes allows them to match more complex features while maintaining precision.

**Quantification** Semantic regexes also make use of the regular expression quantifier zero or one (`?`). As an example, `[:symbol a:][:field color:]?[:field flower:]` matches `a` `red` `rose` and `a` `da` `isy`. Quantifiers allow additional flexibility in the semantic regex.

## 4 METHODS

To study how the semantic regex language alters feature interpretability, we embed it within a standard automated interpretability pipeline (Bills et al., 2023; Paulo et al., 2024; Templeton et al., 2024; Puri et al., 2025). The pipeline consists of three components: a *subject* model whose features we describe, an *explainer* model that generates feature descriptions, and an *evaluator* model that scores descriptions. Given a subject model feature and its activating data, we prompt an explainer model to produce a description in natural or semantic regex language and use the evaluator model to score how well the description matches the feature's behavior. Following prior interpretability work (Bills et al., 2023; Gur-Arieh et al., 2025) and our ablation study in Appendix C, we use `GPT-4o-mini` as the explainer and evaluator. This pipeline allows us to directly compare semantic regexes to natural language descriptions across varied axes of interpretability. It also demonstrates that by decoupling the format of the feature description from the generation process, semantic regexes are compatible with existing and future automated interpretability pipelines.

### 4.1 COLLECTING MODEL FEATURES AND ACTIVATIONS

We study semantic regexes on two families of *subject* models, GPT-2 (Radford et al., 2019) and Gemma-2 (Mesnard et al., 2024). While our approach could describe model neurons (or any component matched to activating data), we focus on describing latent features identified by sparse coding methods (e.g., SAEs) since they often represent monosemantic concepts that are easier to describe and interpret (Bricken et al., 2023).

**GPT-2** We apply semantic regexes to `GPT-2-Small` (Radford et al., 2019) and its residual layer features from Bloom (2024), identified using SAEs with 24,576 features (`GPT-2-RES-25k`).

**Gemma** We also apply semantic regexes to `Gemma-2-2B` (Mesnard et al., 2024) and its Gemma Scope (Lieberum et al., 2024) residual layer features (`Gemma-2-2B-RES-16k` and `Gemma-2-2B-RES-65k`).

To collect activating data for each feature, we use the Neuronpedia (Lin, 2023) API. It provides activating data from OpenWebText (Gokaslan et al., 2019) for `GPT-2-Small` and Pile Uncopyrighted (Gao et al., 2021) for `Gemma-2-2B`.

### 4.2 GENERATING FEATURE DESCRIPTIONS

Given features and their activating data, we prompt an *explainer* model (`GPT-4o-mini`) to generate feature descriptions. We compare our `semantic-regex` method against natural language feature description methods: `token-act-pair` (Bills et al., 2023) and `max-acts` (Paulo et al., 2024). These methods are commonly used and vary in how they generate descriptions, allowing us to get a broad understanding of how semantic regexes compare to natural language descriptions. See Appendix F.1 for implementation details.

**token-act-pair** The `token-act-pair` method is based on Bills et al. (2023) and its Neuronpedia implementation (Lin, 2023) (`oai_token-act-pair`). Here, the explainer model is prompted with the feature description task and three few-shot examples. Given a feature's top five activating examples it is asked to continue the sentence "*the main thing this neuron does is find*". Each example is displayed as a list of tokens and their normalized activation values, like "*these 8\n tokens 10\n activate 0*".

**max-acts** `max-acts` follows Neuronpedia's (Lin, 2023) implementation of Paulo et al. (2024) (`eleuther_acts_top2`). Instead of showing each token's activation value, like in `token-act-pair`, it maintains a more natural text format by delimiting activating tokens within the text, like "*⟨⟨these tokens⟩⟩ activate*". It prompts the explainer model with the feature description task and three few-shot examples. Then, given the feature's top 20 activating examples, the explainer model is asked to "*describe the text latents that are common in the examples*".

**semantic-regex** Since our goal is to accurately describe LLM features, `semantic-regex` only differs from natural language methods in the language available to the explainer model. We simply inject semantic regex specific instructions into existing prompting strategies. To take advantage of its efficient example formatting, we adapt `max-acts`' prompt by updating the instructions to follow the semantic regex language, adding a concise definition of the semantic regex language, and changing the few-shot examples to demonstrate semantic regex primitives and modifiers. We show the explainer model a feature's top 10 activating examples and ask it to "*output a short explanation followed by a semantic regex*", which we find improves the model's ability to follow the syntax.

### 4.3 Evaluating Feature Descriptions

We evaluate feature descriptions using common automated interpretability metrics (Paulo et al., 2024; Puri et al., 2025). Each metric uses an *evaluator* model (`GPT-4o-mini`) to evaluate the descriptions' fidelity to the feature's behavior. *Generation* metrics test the description's ability to generate activating examples (akin to precision), *discrimination* metrics test the description's ability to match known activating examples (akin to recall), and *faithfulness* metrics test the description's ability to match steered generation (a measure of causality). Implementation details in Appendix F.2.

**Generation metrics** Generation metrics evaluate a feature description's ability to generate activating examples. We use `clarity` (Puri et al., 2025), which compares generated and random examples' activations using the Gini index (a rescaling of the ROC AUC). Under this metric, overly broad descriptions score low by generating data outside the feature's activation space, while ideal or slightly narrow descriptions score high by only generating activating data.

**Discrimination metrics** Discrimination metrics test if the features description matches known activating examples. We compute `detection` and `fuzzing` from Paulo et al. (2024) and `responsiveness` and `purity` from Puri et al. (2025). These metrics ask the evaluator model whether the feature description matches activating and random examples. Given the match results, `detection` measures balanced accuracy, `responsiveness` the Gini index, and `purity` average precision. Instead of matching the entire example, `fuzzing` asks if the description matches the example's activating tokens and computes the balanced accuracy of these more specific match results. Under these metrics, overly narrow descriptions score low by missing activating examples, while ideal or slightly broad descriptions score high by covering the entire activation space.

**Faithfulness metrics** Faithfulness tests how well the description reflects causal interventions on the feature. We use `faithfulness` (Puri et al., 2025), which asks the evaluator model whether the feature description matches continuations of random text when the feature is steered versus ablated.

## 5 Results

### 5.1 Semantic Regexes are as Accurate as Natural Language Descriptions

The goal of automated interpretability is to generate feature descriptions that accurately characterize a feature's activations. We benchmark semantic regexes against common natural language description methods (`token-act-pair` and `max-acts`) using discrimination, generation, and faithfulness

Figure 2: Semantic regexes perform on par with natural language feature descriptions across evaluations on `GPT-2-RES-25k`, `Gemma-2-2B-RES-16k`, and `Gemma-2-2B-RES-65k`, suggesting that the semantic regex language is appropriately expressive to describe LLM features.

metrics. To ensure our results generalize across models and features, we evaluate 100 features per layer from `GPT-2-RES-25k`, `Gemma-2-2B-RES-16k`, and `Gemma-2-2B-RES-65k`.

Semantic regex descriptions perform on par with natural language (Figure 2). Specifically, semantic regexes are non-inferior ($p < 0.05$)[1] to natural language on `clarity` across all models, `detection` for `GPT-2-RES-25k`, and `responsiveness` for `Gemma-2-2B-RES-65k`. Moreover, `semantic-regex` outperforms `token-act-pair` ($p < 0.05$)[1] on `clarity` across all models, and `detection`, `fuzzing`, and `responsiveness` on `GPT-2-RES-25k` and `Gemma-2-2B-RES-65k`. This is non-obvious, as the semantic regex language is significantly constrained compared to the grammatical flexibility of natural language. Moreover, while the explainer and evaluator models are well-versed in natural language, they learned the semantic regex language from only a brief description and few-shot examples. These results suggest that imposing structure on feature descriptions does not reduce their accuracy, while offering advantages that we explore in the following sections.

## 5.2 SEMANTIC REGEXES IMPROVE CONCISENESS AND CONSISTENCY

While semantic regexes are similarly accurate to natural language descriptions, their structure offers distinct benefits for interpreting LLM feature behavior (Figure 3).

**Semantic regexes are more concise than natural language descriptions.** Concise feature descriptions adhere to explanation norms (Tim Miller, 2019), making them easier to scan and interpret. While natural language descriptions often require verbose phrases to capture a feature's activation pattern (Huang et al., 2023), semantic regexes can encode the same information in a more compact form using its signal-rich components. For example, in Figure 3 top, the verbose natural language description "*The presence of the sequence 54 indicating a year, time, or numeric reference frequently associated with events*" can be expressed as `[:symbol 54:]`. Quantitatively, semantic regexes are consistently shorter. Across 100 randomly sampled features per layer in `GPT-2-RES-25k`, `Gemma-2-2B-RES-16k`, and `Gemma-2-2B-RES-65k`, the median description length is 41 characters (IQR: 19–59) compared to 139 (IQR: 119–166) for `max-acts` and 55 (IQR: 46–66) for `token-act-pair`. This conciseness directly benefits interpretability, allowing human evaluators in Section 5.4 to parse shorter descriptions without distracting details.

**Semantic regexes are more consistent than natural language descriptions.** LLMs often contain redundant features that activate on similar inputs and serve the same function. In circuit identification applications, recognizing redundant features helps reduce circuit complexity and identify the complete mechanistic circuit (Ameisen et al., 2025). Since the semantic regex language constrains the space of allowable expressions, it produces more consistent descriptions for similar features, making redundancy easier to detect. For example, in Figure 3 middle, two redundant features from different layers of `Gemma-2-2B-RES-16k` both activate on the token "*Advertisement*". While their natural language descriptions differ ("*the word Advertisement*" vs. "*advertisement markers*"), their semantic regex descriptions are identical: `[:symbol Advertisement:]`.

To quantify this effect, we measure consistency by asking how often a method produces the same description when given different random samples of a feature's activating data. This simulates redundancy, since the underlying feature is fixed but observed activating examples vary. Eval-

---

[1] One-sided paired $t$-test with non-inferiority margin $\Delta = 5\%$ and Bonferroni correction for superiority.

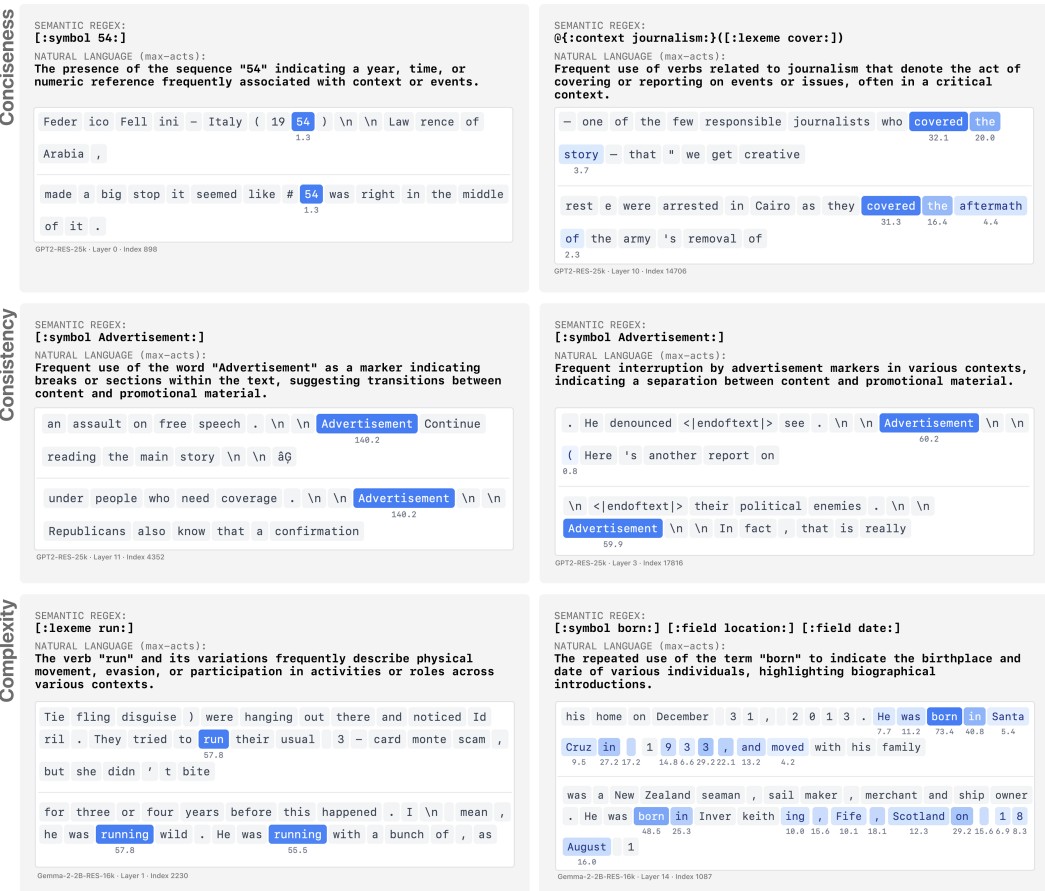

Figure 3: Semantic regexes are often more concise (top), more consistently describe equivalent features (middle), and better reflect feature complexity (bottom) than natural language descriptions.

uating `semantic-regex`, `max-acts`, and `token-act-pair` on five random features per layer of `GPT-2-RES-25k`, each with five generated descriptions, we find that `semantic-regex` yields identical descriptions 33.6% of the time, compared to 12.2% for `token-act-pair` and 0.0% for `max-acts`. These results suggest that constraining the description space with semantic regexes improves consistency, making it easier to detect redundant features.

## 5.3 SEMANTIC REGEXES REFLECT FEATURE COMPLEXITY

Beyond matching the accuracy of natural language descriptions and offering benefits like conciseness and consistency, the structured format of semantic regexes offers additional affordances for model interpretability. Each semantic regex is built from primitives that span levels of abstraction, where symbols match specific characters, lexemes extend to syntactic variants, and fields encode semantic relationships. The number of components in a semantic regex is also a proxy for feature complexity, where features described using a single primitive (e.g., `[:symbol left:]`) are typically conceptually simpler than features that require multiple compositions of primitives and modifiers (e.g., `@{:context political affiliations:}([:symbol left:]|[:symbol right:])`).

We use the level of abstraction encoded in semantic regexes to measure feature complexity, finding that features become more complex deeper in the model (Figure 4). We generate semantic regexes for 1,000 features per layer in `GPT-2-RES-25k`, `Gemma-2-2B-RES-16k`, and `Gemma-2-2B-RES-65k`. While early-layer features are described by smaller and simpler semantic regexes, longer and more abstract semantic regexes are needed to describe later-layer features. In particular, we find that the average number of components per semantic regex (i.e., symbols, lex-

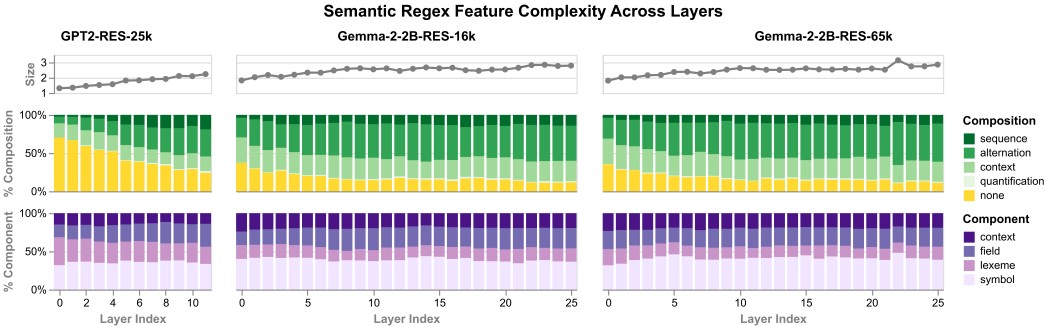

Figure 4: Semantic regexes encode feature complexity. The number (top) and abstraction (middle, bottom) of components increase across model layers, indicating increasingly complex features.

emes, fields, and contexts) increases across layers. This shift is mirrored in the composition of a semantic regex, where early layers have a greater proportion of single-primitive descriptions which decreases in favor of combinations of primitives and modifiers, particularly sequence and alternation compositions. Similarly, we observe that the types of primitives reflect increasing feature complexity. While all semantic regexes contain primitives, we see a decrease in low-level primitives, like lexemes, and an increase in fields (the most abstract primitive). We show an example of increasing feature complexity from early to later layers in Figure 3 bottom.

Together, these trends indicate that later layers require longer and more abstract semantic regexes to capture their complex feature behaviors. This aligns with prior research demonstrating that later layers encode increasingly complex representations (Tenney et al., 2019; Jin et al., 2025; Sun et al., 2025). However, unlike prior methods that rely on model probes or feature testing, we are able to read this complexity directly from the semantic regex feature description. As a result, while like natural language descriptions, semantic regexes allow us understand individual features, their structure also allows us to interpret model-wide attributes.

## 5.4 SEMANTIC REGEXES HELP PEOPLE BUILD MENTAL MODELS OF LLM FEATURES

A common role of feature descriptions is to convey the feature's behavior to a human interpreter (Lin, 2023; Ameisen et al., 2025). Thus, we investigate how semantic regexes impact people's mental models of LLM features (full protocol in Appendix E). We conducted a 24-person study with AI experts representative of people most likely to use feature descriptions in practice. To obtain insights across a range of features, we used 12 `GPT-2-RES-25k` features with diverse activation patterns and accurate natural language and semantic regex descriptions. Given a description, participants were asked to generate three activating phrases and one near-miss counterfactual.

Forming an accurate mental model of an LLM feature means being able to express the feature's decision boundary — i.e., what does and does not activate the feature. We quantify this by measuring the difference between the feature's maximum activation on each participant's positive phrases and their counterfactual phrase. We choose the maximum positive phrase to account for activation artifacts that can result from tokenization. Small or negative values indicate that the participant had difficulty distinguishing activating and non-activating phrases, while large positive values reflect an understanding of the feature's decision boundary. We compare the mean differences of `max-acts` and `semantic-regex` descriptions of the same feature, finding that participants scored as well or higher using semantic regex descriptions on 9 of 12 features (Figure 5).

Although both description types were accurate, natural language often introduced extraneous details that misled participants. For instance, one feature activated on variants of the phrase "*expected to*", but its natural language description included "*'expected to' is frequently used to indicate anticipation or prediction regarding future events or outcomes*". A participant over-indexed on the additional detail, expecting the highly activating phrase "*He does not know meaning of the phrase expected to*" to be a counterexample because it did not indicate anticipation. In contrast, `[:lexeme expect:][:symbol to:]` more concisely expressed the activation pattern (see Section 5.2) and enabled participants to generate strongly activating positives and non-activating counterfactuals.

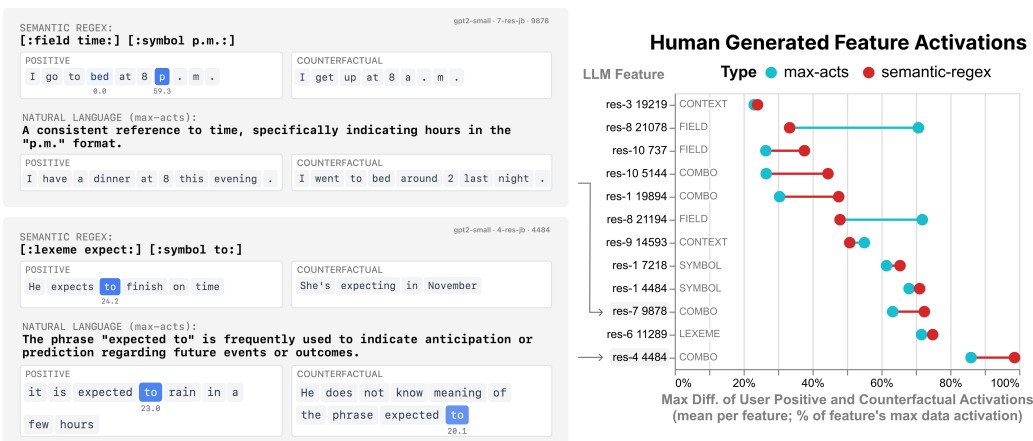

Figure 5: With semantic regexes, user study participants generated strongly activating positive examples and non-activating counterfactuals, indicating their understanding of the feature.

We also find that semantic regexes can reduce ambiguity by conveying activation patterns via example. For a feature that activates on times followed by "*p.m.*", the semantic regex specifies this pattern directly as `[:field time:][:symbol p.m.:]`, and as a result, every participant included "*p.m.*" in their positive phrases. In contrast, natural language descriptions can leave room for interpretation, where one participant misinterpreted "*reference to time, specifically indicating hours in the 'p.m.' format*" as any time in the evening and left off the critical "*p.m.*". A similar pattern occurred in a feature for political conjunctions, where the semantic regex directly included `[:symbol and:]|[:symbol or:]`, making the conjunction pattern unambiguous.

While concise semantic regexes generally benefited participants, there were cases where the verbosity of natural language was more effective. In the two features where natural language substantially outperformed semantic regexes, the semantic regexes were accurate but too minimal to elicit strongly activating phrases. One feature activated on days of the week used within full sentences. Its semantic regex `[:field days of the week:]` led many participants to produce single-word outputs (e.g., "*Monday*") which activated slightly. Whereas, the natural language description "*days of the week indicating specific events*" prompted longer phrases with larger activations.

Finally, we find that participants were able to understand and use semantic regexes with minimal instruction. Despite receiving only a short description and a single example, participants generally interpreted the language correctly. In fact, we received more clarification questions about how to interpret the natural language descriptions than semantic regexes. This contrasts prior work suggesting that structured languages come at a cost because they require "specialized training" (Huang et al., 2023), and instead signals their promise as tools for LLM interpretability.

## 6 DISCUSSION AND LIMITATIONS

Semantic regexes provide a structured syntax for describing LLM features. In doing so, they result in feature descriptions that are more consistent and concise than natural language while still matching its expressive power. However, designing this structured language introduces trade-offs that create both benefits and limitations of semantic regexes and point to several directions for future work.

Since natural language descriptions often contain irrelevant details, we designed semantic regexes to be concise. While this generally helps identify the pertinent activation pattern, it can produce overly terse descriptions. For example, `[:field musician:]` describes a feature that activates on famous musicians like "*Taylor Swift*", incorrectly implying that "*guitarist*" would strongly activate the feature. Striking a balance between conciseness and expressivity across all features may involve adjusting model prompts, extending the language to encode hierarchical concepts (e.g., `[:field musician.name:]`), or using validation loops (Shaham et al., 2024) to mitigate ambiguity.

Additionally, although semantic regexes increase consistency, they do not enforce a unique mapping from activation pattern to description. Many valid semantic regexes can describe the same feature, even if some are less readable, like `@{:context Germany:}([:symbol German:])`. This non-uniqueness is not inherently problematic. In programming languages, for instance, there are many equivalent ways to implement the same function. However, languages often develop style guides that suggest the most readable syntax (van Rossum et al., 2001). Similar heuristics may benefit the semantic regex language, particularly when people (rather than models) are the intended interpreters.

To keep the semantic regex syntax minimal, we leave some components underspecified. For example, `symbols` match exact strings, but the semantic regex language does not specify if they also match string variations (e.g., "*fruit*" and "*Fruit*"). This simplicity avoids an overly complex vocabulary but can cause the model to make inconsistent assumptions across features (e.g., sometimes outputting `[:symbol fruit:]` and other times `[:symbol fruit:]|[:symbol Fruit:]`). Incorporating additional components, like case-insensitive flags, could reduce this ambiguity, especially as growing familiarity with semantic regexes may allow for more complex syntax.

These limitations also impact feature description evaluations. Since current metrics rely on discrete judgments of the feature description, issues like non-uniqueness and ambiguity can lead to false negatives and false positives for both semantic regexes and natural language methods. Developing more expressive metrics, such as continuous scoring schemes or readability evaluations, could provide a more comprehensive understanding of the differences between feature description methods.

Although our results indicate that semantic regexes are expressive, structured, and helpful for human interpretation, additional evaluations could deepen our understanding. For instance, large-scale repetitions of the quantitative analysis could measure pipeline stability and larger crowdsourced user studies could statistically quantify the value of semantic regexes and whether LLM-as-a-judge evaluations faithfully capture human preferences in this domain.

Semantic regexes are not a solution to polysemanticity. While composition helps capture simple polysemantic features (e.g., `[:field clothing:]|[:field Ernest:]`), high degrees of concept entanglement often result in incoherent descriptions, like `@{:context syntax:}([:symbol by:]|[:symbol (:]|[:symbol last:]|[:symbol then:]|...|[:symbol ->:])`. However, since semantic regexes are agnostic to the generation pipeline, they could easily slot into new methods for disentangling polysemantic activations (Kopf et al., 2025).

Semantic regexes require models to learn a novel language from only a brief description. As a result, we observe "grammatical" errors in generation and evaluation (e.g., misinterpreting a primitive). Although models also make mistakes on natural language descriptions, they typically misidentify the activation pattern rather than misunderstand the language. This gap raises questions for automated interpretability. People can learn structured languages and structured languages should better capture the computational roles of features, akin to how programming languages better represent algorithms. However, if models bias towards natural language, structured languages face a barrier to adoption in automated interpretability pipelines. Future work could investigate whether automated interpretability evaluations reflect the needs and abilities of human interpreters.

## 7 CONCLUSION

We introduce semantic regexes, a structured language for automatically describing LLM features. The semantic regex language is grounded in current understanding of LLM features. Each primitive and modifier is designed to reflect patterns observed in interpretability research — i.e., that features often respond to exact tokens (`symbol`), syntactic word forms (`lexeme`), and semantic categories (`field`), and their activations are domain-dependent (`context`) and co-occur with other patterns (Lin, 2023; Templeton et al., 2024). As our understanding of LLM representations grows, we expect the semantic regex language will evolve, with components added or altered to capture new feature behaviors. Just as there are many programming languages to meet different goals, future languages for interpretability may be developed with affordances suited to particular interpretability tasks, such as highlighting input vs. output features, describing particular model components (e.g., attention heads), or exposing safety-relevant features. The affordances of consistency, conciseness, and complexity that we build into semantic regexes expose a broader design space of structured languages for interpretability that can improve our collective understanding and control of LLMs.

ETHICS STATEMENT

As part of this research, we conducted a user study where we surveyed 24 participants from within our institution (Section 5.4). All participants gave informed consent and were informed they could withdraw at any time. We did not collect any identifying information, and all participant responses were anonymized. The study was approved under our institution's internal user survey policies.

REPRODUCIBILITY STATEMENT

Implementation details for the semantic regex method, the baseline methods, and the evaluation metrics are in Appendix F, including full prompts and hyperparameters. Details to recreate our user study, including the survey instructions, are listed in Appendix E. Code is available at `https://github.com/apple/ml-semantic-regex`, and an interactive interface displaying our results is available at `https://apple.github.io/ml-semantic-regex/`.

ACKNOWLEDGMENTS

We thank our colleagues at Apple, with a special mention to Mary Beth Kery, Hadas Kotek, and Lindia Tjuatja, for giving feedback on early drafts of this work.

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

## A   LLM USAGE STATEMENT

Large language models (LLMs) are the subject of this work, and they were also used as general-purpose tools to assist with writing and coding. All research ideas, study designs, analyses, and substantive code implementations were developed by the authors.

## B   ADDITIONAL RESULTS

Here we show alternative views of the benchmarking results from Section 5.1. We show the numerical means and standard deviations in Table A1 and the metric distributions in Figure A1.

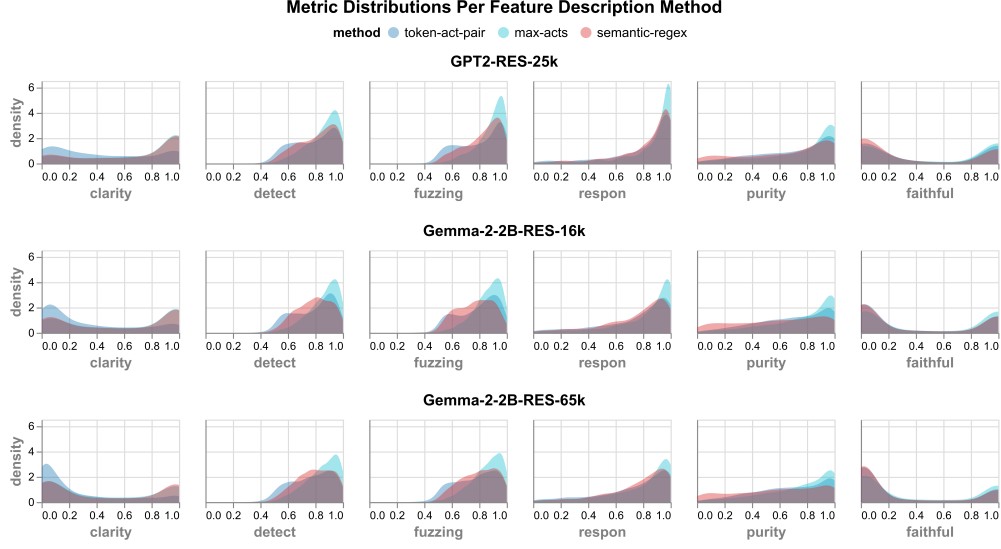

Figure A1: Metric distributions comparing `semantic-regex` feature descriptions against natural language baseline methods. Results are shown for 100 feature per layer on `GPT-2-RES-25k`, `Gemma-2-2B-RES-16k`, and `Gemma-2-2B-RES-65k` and visualized using kernel density estimation with bandwidth estimated via Scott's rule.

Table A1: Across results on `GPT-2-RES-25k`, `Gemma-2-2B-RES-16k`, and `Gemma-2-2B-RES-65k`, `semantic-regex` feature descriptions perform on par with natural language feature description methods. These results suggest that the semantic regex language is appropriately expressive to describe features with similar performance as unconstrained natural language. Each metric is computed on 100 randomly selected features per model layer and displayed as the mean ± the standard deviation.

| | Generation | Discrimination | | | | Faithfulness |
|---|---|---|---|---|---|---|
| **GPT-2-RES-25k** | clarity | detection | fuzzing | responsiveness | purity | faithfulness |
| token-act-pair | $0.45 \pm 0.36$ | $0.79 \pm 0.15$ | $0.80 \pm 0.16$ | $0.81 \pm 0.23$ | $0.71 \pm 0.28$ | $0.46 \pm 0.46$ |
| max-acts | $0.70 \pm 0.35$ | $0.86 \pm 0.12$ | $0.88 \pm 0.10$ | $0.87 \pm 0.20$ | $0.78 \pm 0.26$ | $0.52 \pm 0.46$ |
| semantic-regex | $0.68 \pm 0.36$ | $0.81 \pm 0.14$ | $0.83 \pm 0.13$ | $0.83 \pm 0.21$ | $0.66 \pm 0.32$ | $0.37 \pm 0.45$ |
| | | | | | | |
| **Gemma-2-2B-RES-16k** | | | | | | |
| token-act-pair | $0.34 \pm 0.34$ | $0.78 \pm 0.15$ | $0.79 \pm 0.15$ | $0.76 \pm 0.25$ | $0.69 \pm 0.26$ | $0.38 \pm 0.45$ |
| max-acts | $0.59 \pm 0.39$ | $0.86 \pm 0.11$ | $0.86 \pm 0.10$ | $0.82 \pm 0.22$ | $0.77 \pm 0.24$ | $0.49 \pm 0.46$ |
| semantic-regex | $0.57 \pm 0.40$ | $0.79 \pm 0.13$ | $0.77 \pm 0.13$ | $0.76 \pm 0.23$ | $0.58 \pm 0.30$ | $0.38 \pm 0.45$ |
| | | | | | | |
| **Gemma-2-2B-RES-65k** | | | | | | |
| token-act-pair | $0.25 \pm 0.32$ | $0.77 \pm 0.15$ | $0.77 \pm 0.15$ | $0.75 \pm 0.26$ | $0.69 \pm 0.26$ | $0.30 \pm 0.41$ |
| max-acts | $0.45 \pm 0.40$ | $0.85 \pm 0.12$ | $0.85 \pm 0.12$ | $0.79 \pm 0.23$ | $0.75 \pm 0.25$ | $0.40 \pm 0.44$ |
| semantic-regex | $0.47 \pm 0.42$ | $0.79 \pm 0.13$ | $0.79 \pm 0.13$ | $0.76 \pm 0.22$ | $0.59 \pm 0.31$ | $0.28 \pm 0.40$ |

# C  ABLATION STUDY

Our experiments (Section 5) use `GPT-4o-mini` as both the *explainer* and *evaluator* model. This choice follows common practice in automated interpretability, where the same model is used to generate and evaluate feature descriptions (Bills et al., 2023; Gur-Arieh et al., 2025). Since verification is typically easier than generation, a model can reliably judge its descriptions, and the independence of model calls prevents conditioning effects.

To assess whether our findings depend on the choice of model, we conduct an ablation using a more capable model (`GPT-4o`) in place of `GPT-4o-mini`. This ablation tests whether the relative performance between semantic regexes and natural language descriptions is sensitive to model capability and evaluates the robustness of our results.

Our ablation study follows the same automated interpretability pipeline described in Section 4. All components of the pipeline remain unchanged, including feature extraction, activation formatting, and prompting strategy. The only difference is that `GPT-4o` is substituted for `GPT-4o-mini` in both the explanation and evaluation steps. We apply this setup to a randomly sampled subset of 520 features from across layers of `GPT-2-RES-25k`. For each feature, we generate descriptions using `semantic-regex`, `max-acts`, and `token-act-pair` using both `GPT-4o` and `GPT-4o-mini`, and we evaluate each description using the discrimination and generation metrics used in the main experiments. This design isolates the effect of model capability while preserving all other aspects of the experimental pipeline.

We report the results in Figures A2 and A3 and Table A2. Across all feature description methods and evaluation metrics, the overall results remains the same regardless of whether the explainer and evaluator are `GPT-4o` or `GPT-4o-mini`. Semantic regexes continue to match the performance of natural language descriptions, and the relative differences between description types are nearly identical across the two model settings. While some metrics are slightly higher under `GPT-4o` (e.g., slight improvements in `purity` and `responsiveness`) these gains arise uniformly across methods and are well within the experimental noise.

Overall, the consistency between `GPT-4o-mini` and `GPT-4o` demonstrates that the comparative performance of semantic regexes is not sensitive to the choice of model. Given this robustness, along with the significantly improved efficiency and lower cost of `GPT-4o-mini` (Appendix D), we use `GPT-4o-mini` for all experiments reported in the main text.

Table A2: Across results on `semantic-regex`, `max-acts`, and `token-act-pair` feature descriptions, the choice of explainer and evaluator model (either `GPT-4o` or `GPT-4o-mini`) does not change the relative performance. These results validate our use of `GPT-4o-mini` in our main experiments. Each metric is computed on 520 randomly selected features from `GPT-2-RES-25k` and displayed as the mean $\pm$ the standard deviation.

| | Generation | Discrimination | | | |
|---|---|---|---|---|---|
| **token-act-pair** | clarity | detection | fuzzing | responsiveness | purity |
| GPT-4o-mini | $0.45 \pm 0.37$ | $0.79 \pm 0.15$ | $0.81 \pm 0.15$ | $0.83 \pm 0.22$ | $0.73 \pm 0.26$ |
| GPT-4o | $0.59 \pm 0.37$ | $0.83 \pm 0.13$ | $0.82 \pm 0.13$ | $0.90 \pm 0.17$ | $0.84 \pm 0.21$ |
| | | | | | |
| **max-acts** | | | | | |
| GPT-4o-mini | $0.71 \pm 0.36$ | $0.86 \pm 0.11$ | $0.89 \pm 0.10$ | $0.88 \pm 0.20$ | $0.81 \pm 0.24$ |
| GPT-4o | $0.73 \pm 0.33$ | $0.86 \pm 0.09$ | $0.85 \pm 0.09$ | $0.91 \pm 0.14$ | $0.86 \pm 0.19$ |
| | | | | | |
| **semantic-regex** | | | | | |
| GPT-4o-mini | $0.69 \pm 0.37$ | $0.81 \pm 0.13$ | $0.83 \pm 0.12$ | $0.84 \pm 0.21$ | $0.68 \pm 0.32$ |
| GPT-4o | $0.68 \pm 0.36$ | $0.82 \pm 0.12$ | $0.81 \pm 0.12$ | $0.84 \pm 0.22$ | $0.74 \pm 0.30$ |

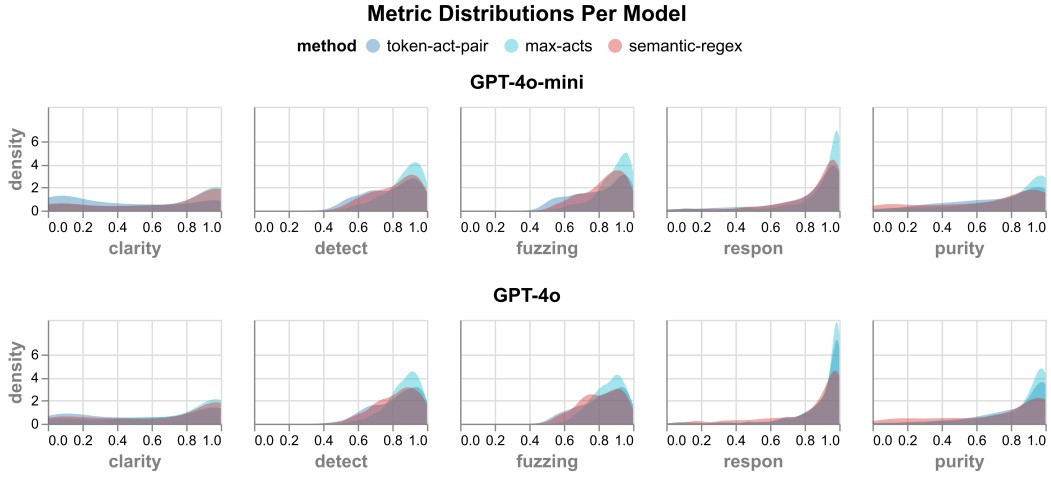

Figure A2: Metric box plots comparing different explainer and evaluator models (`GPT-4o` and `GPT-4o-mini`) across `semantic-regex` and natural language baseline methods (`max-acts` and `token-act-pair`). Results are shown for 520 features from `GPT-2-RES-25k`.

Figure A3: Metric distributions comparing different explainer and evaluator models (`GPT-4o` and `GPT-4o-mini`) across `semantic-regex` and natural language baseline methods. Results are shown for 520 features from `GPT-2-RES-25k` and visualized using kernel density estimation with bandwidth estimated via Scott's rule.

Table A3: Cost analysis for feature description generation. Costs are computed from the number of input tokens (system prompt, few-shot examples, and feature input) and the number of generated output tokens. We report the mean ± std of input and output token counts across 130 features from `GPT-2-RES-25k` (10 per layer). Tokenization is performed using OpenAI's `GPT-4o` model family tokenizer.

| | token-act-pair | max-acts | semantic-regex |
|---|---|---|---|
| Input Tokens: system and few-shot examples ($T_{\text{prompt}}$) | 919 | 483 | 993 |
| Input Tokens: average per feature ($T_{\text{feature}}$) | $457 \pm 213$ | $524 \pm 90$ | $237 \pm 47$ |
| Output Tokens: average per generation ($T_{\text{out}}$) | $9 \pm 3$ | $30 \pm 7$ | $33 \pm 12$ |
| `GPT-4o-mini` cost per feature | $ 0.00021180 | $ 0.00016905 | $ 0.00020430 |
| `GPT-4o` cost per feature | $ 0.00353000 | $ 0.00281750 | $ 0.00340500 |

# D    COST ANALYSIS

The cost of generating `semantic-regex` feature descriptions is comparable to the cost of natural language methods. Computing descriptions for all features in `GPT-2-RES-25k` would cost approximately $65 using `GPT-4o-mini` and $1,088 using `GPT-4o`. These values are on par with existing methods. Generating all `token-act-pair` descriptions would cost $68, and all `max-acts` descriptions would cost $54 using `GPT-4o-mini`.

The description cost per feature depends on the number of input tokens in the prompt, including the system prompt and few-shot examples ($T_{\text{prompt}}$) and the feature's tokens ($T_{\text{feature}}$), and on the number of generated output tokens ($T_{\text{out}}$). It also depends on the API prices for input tokens ($P_{\text{in}}$) and output tokens ($P_{\text{out}}$).

$$\text{Cost per feature} \; = \; P_{\text{in}} \left( T_{\text{prompt}} + T_{\text{feature}} \right) \; + \; P_{\text{out}} \left( T_{\text{out}} \right), \tag{1}$$

We report the token counts and resulting costs for each feature description method in Table A3. Differences across methods are the result of formatting choices and the number of examples shown to the explainer. For instance, `token-act-pair` uses more verbose activation formatting than `max-acts` and `semantic-regex`, and `semantic-regex` uses fewer activating examples than `max-acts` but adds the semantic regex definition to the system prompt.

At the time of experimentation, the OpenAI API pricing[2] is $0.15/$0.60 USD per million input/output tokens for `GPT-4o-mini` and $2.50/$10.00 USD per million input/output tokens for `GPT-4o`. These estimates assume no input caching, so the reported prices are an upper bound on the total cost of feature description generation.

# E    USER STUDY PROTOCOL

Here we provide the full protocol for our user study in Section 5.4.

Since we already evaluate semantic regex reliability across thousands of features in Section 5.1, the purpose of our user study is not to obtain another quantitative reliability estimate. Instead, our goal is to gain qualitative insight into how semantic regexes shape real-world AI experts' interpretation of model features. To do this effectively, we recruit a representative sample of the practitioners most likely to use semantic regexes in practice and present them with a controlled but diverse set of model features. We intentionally keep the study size to 24 experts and 12 features to make the protocol feasible. Importantly, we find that these sample sizes are sufficient to draw insights about the benefits and limitations of semantic regexes.

**Participants**  Our target population consists of practitioners who would realistically interpret semantic regex feature descriptions in their real-world tasks, including interpretability research, AI safety, and model development. Therefore, we recruit 24 AI experts from within our organization who work with LLMs. These participants have a broad range of technical roles and backgrounds, ensuring that the study captures how semantic regexes are interpreted by the types of experts who would use them in practice.

**Features**  Our user study investigates how the *format* of a feature description influences people's understanding. To ensure that participants interpret meaningful feature descriptions, we restricted to features from `GPT-2-RES-25k` with both accurate semantic regex and natural language descriptions. Because our goal is not to exhaustively evaluate all such features, but rather to study human interpretation across a representative sample, we focus on selecting a diverse subset.

We first filter to features whose `semantic-regex` and `max-acts` descriptions each achieve a `detection` score above 75%. From this filtered set, we manually select 12 features whose descriptions we verified to be accurate. Our selection process emphasizes feature diversity. We select features from across the model's layers, resulting in 5 features from early layers (1-4), 4 from middle layers (5-8), and 3 from late layers (9-12). And, we chose features that represent diverse activation patterns, such that 2 activate on exact `symbols`, 1 on a `lexeme`, 3 on a `field`, 2 only activate in a particular `context`, and 4 were complex patterns that require concatenation or combination of semantic regex components.

---

[2]`https://platform.openai.com/docs/pricing`

This selection process allows us to study human interpretation across a controlled but diverse range of features without overburdening participants. We show all selected features and their descriptions in Figure A4.

**Task**  Participants were tasked with writing phrases that match or were a counterfactual to a given feature description. Each participant completed the task for three natural language feature descriptions (`max-acts`) and three `semantic-regex` feature descriptions. We showed each participant features of varying complexity, with half the participants seeing the feature descriptions in the left column of Figure A4 and the other half seeing feature descriptions from the right column. To avoid carryover effects, participants saw natural language and semantic regex descriptions from different features. To avoid learning effects, we randomized the order, with half the participants seeing natural language descriptions first and the other half seeing semantic regexes first. This process resulted in 425 positive and 143 counterfactual phrases (some participants did not generate all phrases).

The task was completed asynchronously and distributed via a survey link. The survey began with an introduction to the task, the semantic regex language, and an example of matching and counterfactual phrases (Figure A5). Then, participants were shown an explanation of the feature description type they would see first, followed by three feature descriptions they were asked to generate phrases for. This was repeated with the alternate feature description type and the final three explanations (Figures A6 and A7). For each feature description, they were asked to "*Write 3 phrases/sentences that match the feature description*" and "*Write 1 phrase/sentence that is a counterfactual for the semantic regex feature description*" (Figures A8 and A9).

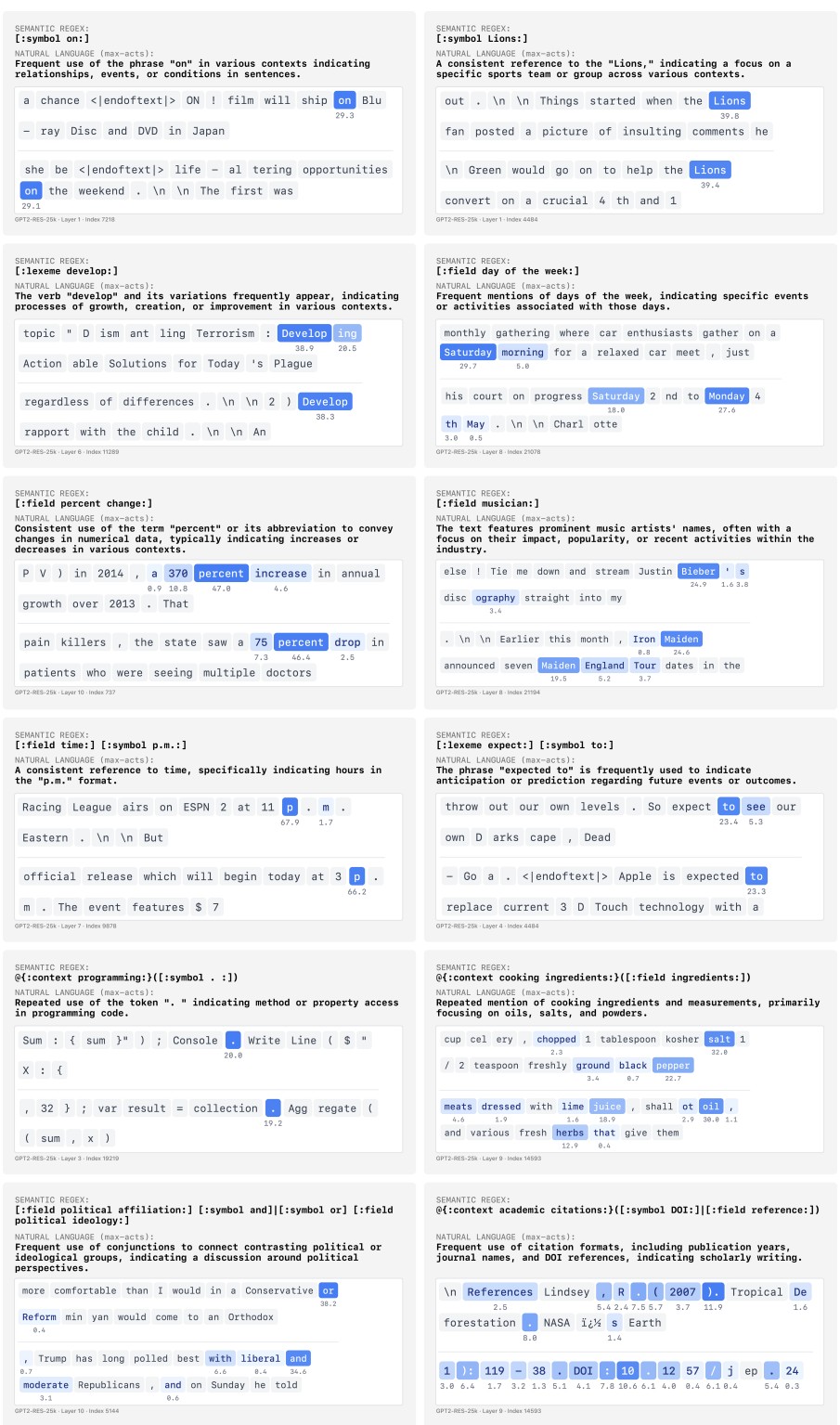

Figure A4: The features we used in our user study (Section 5.4), displayed as their top activating examples and `semantic-regex` and `max-acts` feature descriptions. Half of the participants were shown the features in the left column and the other half was shown features in the right column.

## Welcome!

This survey is part of a research project on LLM feature descriptions. It should take **~10 minutes**. Thank you for taking part!

**Task Overview**

Your job is to write phrases/sentences based on LLM feature descriptions.

For each feature description, you will write:
- 3 phrases/sentences that **match** the description. The goal is that these phrases activate the LLM feature.
- 1 phrase/sentence that is a **counterfactual** for the description (i.e., it almost matches the description but doesn't). The goal is that these phrases do not activate the LLM feature.

You will get **3 natural language** feature descriptions and **3 semantic regex** feature descriptions.

**Semantic Regex Format**

A **semantic regex** is a structured language pattern.

Basic Elements:
- **[:symbol X:]**→ matches X
  - [:symbol color:] matches "image in color"
- **[:lexeme X:]**→ matches X and its syntactic variants
  - [:lexeme color:] matches *color*, *colors*, *coloring*, *colored*, etc. such as "coloring book" and "many colors"
- **[:field X:]** → matches semantic variants of X
  - [:field color:] matches *red*, *orange*, *blue*, etc. such as "blue skies"

These elements can be combined together:
- **@{:context X}(A)**→ matches semantic regex A when it appears in context X
  - @{:context politics:}([:field color:]) matches "a blue state" but not "blue skies"
- **[A][B]**→ matches semantic regex A immediately before semantic regex B
  - [:lexeme color:][:field color:] matches "the color blue" and "colored yellow"
- **[A]|[B]**→ matches semantic regex A OR semantic regex B
  - [:field color:] [:symbol and:]|[:symbol or:] [:field color:] matches "red or blue" or "pink and purple"

**Example**

Natural Language Feature Description: Mentions of time durations associated with prison sentences, highlighting various lengths ranging from months to years.
Semantic Regex Feature Description: @{:context prison sentences}([:field duration:])

Matching Phrases:
1. sentenced to 30 years in prison
2. spent four years behind bars
3. 7 months of jail time

Counterfactual Phrase:
1. spent four years living in California

Your responses are voluntary and you may stop at any time.

Figure A5: The user study introduction explains the task, the semantic regex language, and provides an example.

You will start with **3 semantic regex** feature descriptions. As a reminder:

A **semantic regex** is a structured language pattern.
Basic Elements:
- **[:symbol X:]**→ matches X
  - [:symbol color:] matches "image in color"
- **[:lexeme X:]**→ matches X and its syntactic variants
  - [:lexeme color:] matches *color*, *colors*, *coloring*, *colored*, etc. such as "coloring book" and "many colors"
- **[:field X:]**   → matches semantic variants of X
  - [:field color:] matches *red*, *orange*, *blue*, etc. such as "blue skies"

These elements can be combined together:
- **@{:context X}(A)**→ matches semantic regex A when it appears in context X
  - @{:context politics:}([:field color:]) matches "a blue state" but not "blue skies"
- **[A][B]**→ matches semantic regex A immediately before semantic regex B
  - [:lexeme color:][:field color:] matches "the color blue" and "colored yellow"
- **[A]|[B]**→ matches semantic regex A OR semantic regex B
  - [:field color:] [:symbol and:]|[:symbol or:] [:field color:] matches "red or blue" or "pink and purple"

**Example**
Semantic Regex Feature Description:
@{:context prison sentences}([:field duration:])

Matching Phrases:
 1. sentenced to 30 years in prison
 2. spent four years behind bars
 3. 7 months of jail time

Counterfactual Phrase:
 1. spent four years living in California

Figure A6: The user study's description of semantic regexes.

Now you will see **3 natural language** feature descriptions.

The feature descriptions describe data that activates an LLM feature. Aim to generate phrases/sentences where that description would (or would not) apply.

**Example**
Natural Language Feature Description:
Mentions of time durations associated with prison sentences, highlighting various lengths ranging from months to years.

Matching Phrases:
 1. sentenced to 30 years in prison
 2. spent four years behind bars
 3. 7 months of jail time

Counterfactual Phrase:
 1. spent four years living in California

Figure A7: The user study's description of natural language feature descriptions.

ⓘ * Write 3 phrases/sentences that **match** the *semantic regex* feature description:
    **[:symbol on:]**

ⓘ * Write 1 phrase/sentence that is a **counterfactual** for the *semantic regex* feature description:
    **[:symbol on:]**

*Instructions Reminder:*

A **semantic regex** is a structured language pattern.
Basic Elements:
- **[:symbol X:]**→ matches X
  - [:symbol color:] matches "image in color"
- **[:lexeme X:]**→ matches X and its syntactic variants
  - [:lexeme color:] matches *color*, *colors*, *coloring*, *colored*, etc. such as "coloring book" and "many colors"
- **[:field X:]** → matches semantic variants of X
  - [:field color:] matches *red*, *orange*, *blue*, etc. such as "blue skies"

These elements can be combined together:
- **@{:context X}(A)**→ matches semantic regex A when it appears in context X
  - @{:context politics:}([:field color:]) matches "a blue state" but not "blue skies"
- **[A][B]**→ matches semantic regex A immediately before semantic regex B
  - [:lexeme color:][:field color:] matches "the color blue" and "colored yellow"
- **[A]|[B]**→ matches semantic regex A OR semantic regex B
  - [:field color:] [:symbol and:]|[:symbol or:] [:field color:] matches "red or blue" or "pink and purple"

**Example**
Semantic Regex Feature Description:
@{:context prison sentences}([:field duration:])

Matching Phrases:
1. sentenced to 30 years in prison
2. spent four years behind bars
3. 7 months of jail time

Counterfactual Phrase:
1. spent four years living in California

Figure A8: An example of a user study question for a semantic regex feature description.

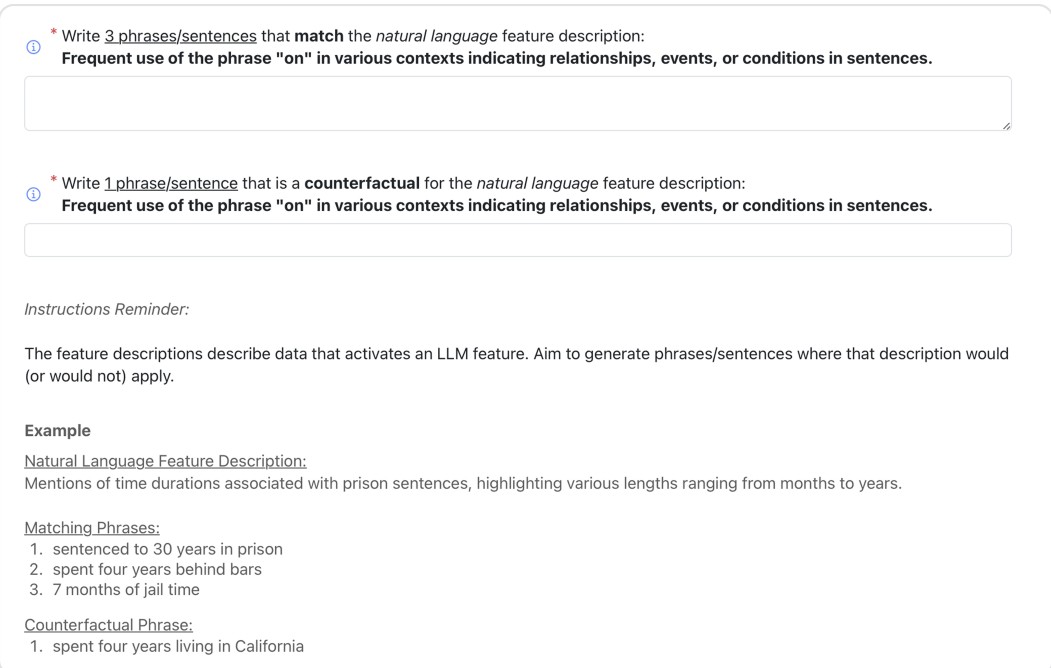

Figure A9: An example of a user study question for a natural language feature description.

Table A4: Hyperparameters used to generate feature descriptions using `semantic-regex` and the natural language benchmarks (`token-act-pair` (Bills et al., 2023) and `max-acts` (Paulo et al., 2024)). Benchmark hyperparameters follow Neuronpedia's implementations of the original methods (Lin, 2023).

| Parameter | token-act-pair | max-acts | semantic-regex |
|---|---|---|---|
| Number of few shot examples | 3 | 3 | 4 |
| Number of data examples | 5 | 20 | 10 |
| Number of tokens per example | 64 | 32 | 32 |
| Explanation model temperature | 1.0 | 0.7 | 1.0 |
| Explanation model top-p | 1.0 | 1.0 | 1.0 |
| Activation threshold | — | 60% | 30% |

# F  IMPLEMENTATION DETAILS

## F.1  FEATURE DESCRIPTION METHODS

We evaluate three approaches for generating descriptions of LLM features: `token-act-pair` (Bills et al., 2023), `max-acts` (Paulo et al., 2024), and our proposed `semantic-regex` method. All methods use `gpt-4o-mini` as the explainer model. We follow Neuronpedia's reference implementations (Lin, 2023) for each baseline method. Implementation details for each method are provided below, and a summary of the settings is shown in Table A4.

**token-act-pair**  We follow Neuronpedia's implementation (Lin, 2023) of the feature description method proposed in Bills et al. (2023), called `oai_token-act-pair` in the Neuronpedia interface. The explainer model is prompted with the original feature description system instructions and the three few-shot examples used in the original paper (Listing 2). For each feature, we supply the top five activating data examples from Neuronpedia (Lin, 2023), where each example consists of up to a 64-token window centered on the maximally activating token. Tokens are presented in the format:

```
<start>
token_1  activation_1
token_2  activation_2
...
<end>
```

where `activation_i` is the normalized activation value for the feature. Activations are linearly scaled between 0 and 10, such that the feature's maximum activation value maps to 10. The prompt ends with the continuation cue "*the main thing this neuron does is find*", following the original protocol. Generation is performed with temperature = 1 and top-p = 1.

**max-acts**  We follow Neuronpedia's implementation (Lin, 2023) of the feature description method proposed in Paulo et al. (2024), called `eleuther_acts_top20` in the Neuronpedia interface. The explainer model is prompted with the original system instructions and the three few-shot examples provided in the original paper (Listing 3). For each feature, we supply the top 20 activating data examples from Neuronpedia (Lin, 2023), where each example consists of up to a 32-token window centered around the maximally activating token. Tokens exceeding 60% of the feature's maximum activation are highlighted and contiguous activating tokens are merged into a single highlighted segment, like:

```
token_1<<token_2token_3>>token4...
```

The prompt ends with the instruction to "*describe the text latents that are common in the examples*". Generation is performed with temperature = 0.7 and top-p = 1, following Neuronpedia's settings.

**semantic-regex**  To generate semantic regexes, we prompt the explainer model with system instructions adapted from `max-acts`, augmented with a concise definition of the semantic regex language (Listing 1) and four few-shot examples illustrating the range of available primitives and

modifiers (Listing 4). For each feature, we supply the top 10 activating data examples from Neuronpedia (Lin, 2023), where each example consists of up to a 32-token window centered on the maximally activating token. Tokens exceeding 30% of the feature's maximum activation are highlighted, and contiguous spans are merged into single segments, as in `max-acts`.

```
token_1<<token_2token_3>>token4...
```

We found this lower threshold produced more consistent highlights, which is important for mapping activations to the structured semantic regex language. The prompt concludes with the instruction to "*output a short explanation followed by a semantic regex*", which we found improved syntactic adherence. Generation is performed with temperature = 1 and top-p = 1.

```
A Semantic Regex is a structured pattern composed of:
* [:symbol X:] - matches an exact phrase X (e.g., [:symbol running:] matches "I am<< running>>" and "<<
running>> faster").
* [:lexeme X:] - matches a phrase X and its syntactic variants (e.g., [:lexeme run:] matches "she<< ran>>",
 "it's<< running>> quickly").
* [:field X:] - matches a phrase X and its semantic variants (e.g., [:field run:] matches "out for a << jog
>>" and "<< sprint>> for gold").
X can be a subword (e.g., ing), word (e.g., running), or phrase (e.g., running tempo).
These components can be combined to match more complex patterns:
* S1 S2 - matches a sequence where S1 is followed by S2 (e.g., [:symbol run:] [:lexeme fast:] matches "I<<
run fast>>" and "they<< run faster>>").
* S1|S2 - matches either S1 or S2 (e.g., [:symbol run:]|[:symbol walk:] matches "I<< run>>" and "I<< walk
>>").
* S? - matches S or nothing (e.g., [:lexeme run:] [:symbol very:]? [:symbol fast:] matches "I am<< running
fast>>" and "I<< run very fast>>").
* @{:context C:}(S) - matches S that only activates in the context C (e.g., @{:context political:}([:lexeme
 run:]) matches "she<< ran>> for office" and "<<running>> for governor" but not "I<< run>> marathons").
```

Listing 1: The `semantic-regex` description that we inject into method and metric prompts.

```
<system>
We're studying neurons in a neural network. Each neuron looks for some particular thing in a short document
. Look at the parts of the document the neuron activates for and summarize in a single sentence what the
neuron is looking for. Don't list examples of words.

The activation format is token<tab>activation. Activation values range from 0 to 10. A neuron finding what
it's looking for is represented by a non-zero activation value. The higher the activation value, the
stronger the match.
</system>

<user>
Neuron 1
Activations:
<start>
t 0
urt 0
ur  0
ro  0
 is 0
 fab  0
ulously 0
 funny  0
 and  0
 over 0
 the  0
 top  0
 as 0
 a  0
 '  0
very  0
 sneaky 0
' 1
 but  0
ler 0
 who  0
 excel  0
s 0
 in 0
 the  0
 art  0
 of 0
 impossible 0
 disappearing 6
/ 0
re  0
app 0
earing  10
 acts 0
<end>
<start>
esc 0
 aping 9
```

```
 the  4
 studio 0
 ,   0
 pic  0
col 0
i 0
 is 0
 warm 0
ly  0
 affecting  3
 and  0
 so 0
 is 0
 this 0
 ad 0
roit  0
ly  0
 minimalist 0
 movie  0
 .  0
<end>

Same activations, but with all zeros filtered out:
<start>
' 1
 disappearing 6
earing  10
<end>
<start>
aping 9
 the  4
 affecting  3
<end>

Explanation of neuron 1 behavior: the main thing this neuron does is find
</user>

<assistant>
present tense verbs ending in 'ing'.
</assistant>

<user>
Neuron 2
Activations:
<start>
as  0
 sac  0
char  0
ine 0
 movies 0
 go 0
 ,  0
 this 0
 is 0
 likely 0
 to 0
 cause  0
 massive  0
 cardiac  0
 arrest 10
 if 0
 taken  0
 in 0
 large  0
 doses  0
 .  0
<end>
<start>
shot  0
 perhaps  0
 '  0
art 0
istically 0
' 0
 with 0
 handheld 0
 cameras  0
 and  0
 apparently 0
 no 0
 movie  0
 lights 0
 by 0
 jo 0
aquin 0
 b  0
aca 0
- 0
as  0
ay  0
 ,  0
 the  0
```

```
 low  0
 - 0
budget  0
 production 0
 swings 0
 annoy  0
ingly 0
 between  0
 vert 0
igo 9
 and  0
 opacity  0
 .  0
<end>

Same activations, but with all zeros filtered out:
<start>
 arrest 10
<end>
<start>
igo 9
<end>

Explanation of neuron 2 behavior: the main thing this neuron does is find
</user>

<assistant>
words related to physical medical conditions.
</assistant>

<user>
Neuron 3
Activations:
<start>
the 0
 sense  0
 of 0
 together 3
ness  7
 in 0
 our  0
 town 1
 is 0
 strong 0
 .  0
<end>
<start>
a 0
 buoy 0
ant 0
 romantic 0
 comedy 0
 about  0
 friendship 0
 ,  0
 love 0
 ,  0
 and  0
 the  0
 truth  0
 that 0
 we 2
're 4
 all  3
 in 7
 this 10
 together 5
 .  0
<end>

Explanation of neuron 3 behavior: the main thing this neuron does is find
</user>

<assistant>
phrases related to community.
</assistant>
```

Listing 2: The `token-act-pair` prompt and few-shot examples. `<system></system>`, `<user></user>`, and `<assistant></assistant>` delineate the message roles passed to the explainer model.

```
<system>
You are a meticulous AI researcher conducting an important investigation into patterns found in language.
Your task is to analyze text and provide an explanation that thoroughly encapsulates possible patterns
found in it.
Guidelines:
```

```
You will be given a list of text examples on which special words are selected and between delimiters like
<<this>>. If a sequence of consecutive tokens all are important, the entire sequence of tokens will be
contained between delimiters <<just like this>>.

- Try to produce a concise final description. Simply describe the text latents that are common in the
examples, and what patterns you found.
- If the examples are uninformative, you don't need to mention them. Don't focus on giving examples of
important tokens, but try to summarize the patterns found in the examples.
- Do not mention the marker tokens (<< >>) in your explanation.
- Do not make lists of possible explanations. Keep your explanations short and concise.
- The last line of your response must be the formatted explanation, using [EXPLANATION]:
</system>

<user>
Example 1:  and he was <<over the moon>> to find
Example 2:  we'll be laughing <<till the cows come home>>! Pro
Example 3:  thought Scotland was boring, but really there's more <<than meets the eye>>! I'd
</user>

<assistant>
[EXPLANATION]: Common idioms in text conveying positive sentiment.
</assistant>

<user>
Example 1:  a river is wide but the ocean is wid<<er>>. The ocean
Example 2:  every year you get tall<<er>>," she
Example 3:  the hole was small<<er>> but deep<<er>> than the
</user>

<assistant>
[EXPLANATION]: The token "er" at the end of a comparative adjective describing size.
</assistant>

<user>
Example 1:  something happening inside my <<house>>", he
Example 2:  presumably was always contained in <<a box>>", according
Example 3:  people were coming into the <<smoking area>>".

However he
Example 4:  Patrick: "why are you getting in the << way?>>" Later,
</user>

<assistant>
[EXPLANATION]: Nouns representing a distinct object that contains something, sometimes preceding a
quotation mark.
</assistant>
```

Listing 3: The `max-acts` prompt and few-shot examples. `<system></system>`, `<user></user>`, and `<assistant></assistant>` delineate the message roles passed to the explainer model.

```
<system>
You are interpreting the role of LLM features. Your task it to describe patterns across activating text
examples.

Input:
You will be given a list of text examples.
Activating phrases in each example are highlighted between delimiters like<< this and that>>.

Output:
You will output a **Semantic Regex** that describes patterns across the text examples.
A Semantic Regex is a structured pattern composed of:
* [:symbol X:] - matches an exact phrase X (e.g., [:symbol running:] matches "I am<< running>>" and "<<
running>> faster").
* [:lexeme X:] - matches a phrase X and its syntactic variants (e.g., [:lexeme run:] matches "she<< ran>>",
 "it's<< running>> quickly").
* [:field X:] - matches a phrase X and its semantic variants (e.g., [:field run:] matches "out for a << jog
 >>" and "<< sprint>> for gold").
X can be a subword (e.g., ing), word (e.g., running), or phrase (e.g., running tempo).
These components can be combined to match more complex patterns:
* S1 S2 - matches a sequence where S1 is followed by S2 (e.g., [:symbol run:] [:lexeme fast:] matches "I<<
run fast>>" and "they<< run faster>>").
* S1|S2 - matches either S1 or S2 (e.g., [:symbol run:]|[:symbol walk:] matches "I<< run>>" and "I<< walk
>>").
* S? - matches S or nothing (e.g., [:lexeme run:] [:symbol very:]? [:symbol fast:] matches "I am<< running
fast>>" and "I<< run very fast>>").
* @{:context C:}(S) - matches S that only activates in the context C (e.g., @{:context political:}([:lexeme
 run:]) matches "she<< ran>> for office" and "<<running>> for governor" but not "I<< run>> marathons").

Instructions:
1. Look at the text examples to identify patterns that occur across **all** examples.
2. First, look for patterns within the << >> delimiters.
    1. If you find an exact phrase, use a [:symbol X:].
    2. If you find a phrase and its syntactic variants, use a [:lexeme X:].
    3. If you find a phrase and its semantic variants, use a [:field X:].
    4. Create a Semantic Regex (S) with the fewest components that precisely describes the pattern.
3. Next, look for patterns in the examples' topics.
```

```
    1. If all examples are related to the same topic (C) AND the topic is not redundant with the current
    Semantic Regex (S), use a @{:context C:}(S) modifier.
4. Output the simplest and most concise Semantic Regex that precisely describes the patterns across all
examples.
5. Do not include the delimiters tokens (<< >>) in your Semantic Regex.
6. Output a short explanation followed by "SR: " and then the Semantic Regex
</system>

<user>
1: ax=[figg.add_subplot(2,1,k+)<< for>> k in xrange(2)]
2: p = 0<< for>> q in qlist: pprev = p
3: << for>> lam, prob in suite.Items():
</user>

<assistant>
The phrase 'for' activates only in the context of coding. SR: @{:context coding:}([:symbol for:])
</assistant>

<user>
1: extradition legislation<< prohibits an individual from being sent back>>
2: << whether Israel can lift the collective protection of asylum seekers>>
3: << called on Dutch>> authorities<< to do more to protect human rights>> workers
</user>

<assistant>
The activating phrases and their surrounding phrases are all related to human rights legislation. SR: [:
field human rights legislation:]
</assistant>

<user>
1: off the markets - 2<< times the size>> of India's
2: needed to meet demand - seven times<< the existing>> number
3: That's about three times<< the>> rate seen in Hawaii
</user>

<assistant>
The symbol 'the' activates when it is between a numerical multiplier and an amount of something. SR: [:
field number:] [:symbol times the] [:field amount:]
</assistant>

<user>
1: sentenced to<< 30 years in>> prison
2: to<< ten months>> in prison suspended for<< 3 years>>
3: << 1 month>> behind bars but was ordered to
</user>

<assistant>
The activating phrases are durations of prison sentences. SR: @{:context prison sentences}([:field duration
:])
</assistant>
```

Listing 4: The `semantic-regex` prompt and few-shot examples. `<system></system>`, `<user></user>`, and `<assistant></assistant>` delineate the message roles passed to the explainer model.

## F.2    EVALUATION METRICS

We evaluate the feature description metrics across a suite of evaluation metrics. We take these evaluation metrics from Paulo et al. (2024) (Eleuther) and Puri et al. (2025) (FADE). We follow their implementations closely, but make some modifications for consistency across metrics. To account for the semantic regex structure, we make small adjustments to the original prompts and inject a description of the semantic regex language (full prompts in Listings 5, 6, 7, 8, 9, 10, 11 and 12). The hyperparameters are listed in Table A5.

All metrics use `gpt-4o-mini` as the evaluator model. Metrics typically rely on a set of activating examples (positives) and random examples (negatives). Unless otherwise specified, we use 50 activating examples and 50 negative examples per feature, each consisting of a 32-token window centered on the maximally activating token. Activating example sampling varies per metric, but all random examples are sampled from alternative features in Neuronpedia; given the number and diversity of features, this approximates random dataset sampling. To evaluate the description, many of the methods compute the feature's activation on these data sets. Following prior work (Paulo et al., 2024), we ignore the beginning-of-sequence tokens when computing activations.

### F.2.1    ELEUTHER METRICS

We implement the `detection` and `fuzzing` metrics from Paulo et al. (2024), following Neuronpedia's implementation (`eleuther_recall` and `eleuther_fuzz`). For Eleuther metrics, activating

Table A5: The hyperparameters used to evaluate feature descriptions using metrics from Paulo et al. (2024) (`detection`, `fuzzing`) and Puri et al. (2025) (`clarity`, `responsiveness`, `purity`, `faithfulness`).

| Parameter | clarity | detection | fuzzing | responsiveness | purity | faithfulness |
|---|---|---|---|---|---|---|
| Number of positive examples | 50 | 50 | 50 | 50 | 50 | — |
| Number of random examples | 50 | 50 | 50 | 50 | 50 | 10 |
| Number of tokens per example | 32 | 32 | 32 | 32 | 32 | 32 |
| Number of examples per model call | — | 5 | 5 | 15 | 15 | 15 |
| Positive example sampling method | — | quantiles | quantiles | percentile | percentile | — |
| Number of quantiles | — | 10 | 10 | — | — | — |
| Percentiles | — | — | — | 0, 50, 75, 95, 100 | 0, 50, 75, 95, 100 | — |
| Top sampling percentage | — | — | — | 20% | 20% | — |
| Evaluation model temperature | 1.0 | 0.7 | 0.7 | 1.0 | 1.0 | 1.0 |
| Number of generation runs | 10 | — | — | — | — | — |
| Modification factors | — | — | — | — | — | 0, 1, 10, 100 |
| Number of steered generation tokens | — | — | — | — | — | 30 |

examples are sampled uniformly across 10 activation quantiles. Across both metrics the evaluator model is tasked with providing a binary judgment of whether the each example matches the feature description. The evaluator model processes 5 examples per call, with temperature = 0.7 and a maximum of 500 completion tokens.

**detection** The `detection` metric evaluates description quality at the example level. The evaluator model prompts are shown in Listings 5 and 6. The evaluator is shown a feature description and an example and asked whether the description matches the example's text. The final score is the balanced accuracy of these binary judgments compared to the ground-truth example labels (positive = activating, negative = random).

**fuzzing** The `fuzzing` metric follows the same setup as `detection` but evaluates matches at the activation level. In each of the examples, we highlight activating tokens, and the evaluator model is asked whether the description matches the highlighted regions. The evaluator model prompts are shown in Listings 7 and 8. For activating examples, we highlight tokens that activate higher than the activation threshold (60% of the feature's maximum activation for `max-acts` and 30% of the feature's maximum activation for `semantic-regex`) and merge contiguously activating tokens, e.g.:

```
token_1<<token_2token_3>>token_4...
```

Random examples are highlighted using the activation pattern of the positive examples, which ensures an equal distribution of activating tokens across both example sets. The evaluator is asked whether the description matches the highlighted regions, and the score is computed as the balanced accuracy of these judgments relative to the ground-truth labels. This yields a stricter variant of `detection` by focusing the match decision on activating regions.

### F.2.2 FADE METRICS

We implement the `clarity`, `responsiveness`, `purity`, and `faithfulness` metrics from Puri et al. (2025), adapting their setup for consistency with the Eleuther metrics. We sample activating examples following FADE's stratified sampling protocol: 20% from the top activations and the remainder sampled uniformly across the percentile bins [0, 50), [50, 75), [75, 95), and [95, 100].

**responsiveness and purity** To compute `responsiveness` and `purity`, the evaluator model is shown a feature description and a data example and asked to provide a discrete judgment (0, 1, or 2) indicating how strongly the text matches the description. Following FADE, samples labeled as 1 are discarded before scoring. The evaluator model prompts are shown in Listings 9 and 10. We parallelize scoring, such that the evaluator judges 15 examples at once. `responsiveness` computes the Gini index of the resulting scores, capturing how well the description's matches rank-order the data by activation strength. `purity` computes the average precision, capturing how well the description separates activating from non-activating examples in a retrieval setting.

**clarity** Unlike the other metrics, `clarity` evaluates a feature description by testing whether it can generate activating text. Here, the evaluator model is prompted with the description and

asked to generate candidate examples (full prompts are shown in Listings 11 and 12). We make 10 independent generation calls, producing a set of positives, and sample an equal number of random examples as negatives. Each example is then passed through the subject model to obtain its activation value. `clarity` is computed as the Gini index, measuring whether generated examples achieve higher activations than the random negatives.

**faithfulness** `faithfulness` measures how well the feature description captures the feature's causal role in the LLM. Unlike the other metrics that are based on analyzing activating data, `faithfulness` compares the feature description against steered model generations where the feature is amplified or ablated. Here, we sample 10 random examples. Then we have the model continue each example by generating 30 additional tokens. We apply this strategy multiple times, where each time the feature is ablated (set to 0) or amplified. In the amplified settings, we set its strength equal to its maximum known data activation multiplied by a modification factor (1, 10, 100). Given the steered generations and the feature description, we follow the same rating task as in `responsiveness` and `purity`, asking the evaluator model to provide a discrete judgment (0, 1, or 2) indicating how strongly the text matches the description. Following FADE, samples labeled as 1 are discarded before scoring. The evaluator model prompts are shown in Listings 9 and 10. We compute faithfulness as the maximum proportion of matching generations at each modification factor compared to the proportion of matching generations when the feature is ablated.

```
<system>
You are an intelligent and meticulous linguistics researcher.

You will be given a certain latent of text, such as "male pronouns" or "text with negative sentiment".

You will then be given several text examples. Your task is to determine which examples possess the latent.

For each example in turn, return 1 if the sentence is correctly labeled or 0 if the tokens are mislabeled.
You must return your response in a valid Python list. Do not return anything else besides a Python list.
</system>

<user>
Latent explanation: Words related to American football positions, specifically the tight end position.

Test examples:
Example 0:<|endoftext|>Getty Images\n\nPatriots tight end Rob Gronkowski had his boss'
Example 1: names of months used in The Lord of the Rings:\n\n"...the
Example 2: Media Day 2015\n\nLSU defensive end Isaiah Washington (94) speaks to the
Example 3: shown, is generally not eligible for ads. For example, videos about recent tragedies,
Example 4: line, with the left side – namely tackle Byron Bell at tackle and guard Amini
</user>

<assitant>
[1,0,1,0,1]
</assistant>

<user>
Latent explanation: The word 'guys' in the phrase 'you guys'.

Test examples:
Example 0: if you are<< comfortable>> with it. You<< guys>> support me in many other ways already and
Example 1: birth control access<|endoftext|> but I assure you<< women>> in Kentucky aren't laughing as they
 struggle
Example 2:'s gig! I hope you guys<< LOVE>> her, and<< please>> be nice,
Example 3:American, told<< Hannity>> that "you<< guys>> are playing the race card."
Example 4:<< the>><|endoftext|>I want to<< remind>> you all that 10 days ago (director Massimil
</user>

<assistant>
[0,0,0,0,0]
</assistant>

<user>
Latent explanation: "of" before words that start with a capital letter.

Test examples:
Example 0: climate, Tomblin's Chief of Staff Charlie Lorensen said.\n
Example 1: no wonderworking relics, no true Body and Blood of Christ, no true Baptism
Example 2: Deborah Sathe, Head of Talent Development and Production at Film London,
Example 3: It has been devised by Director of Public Prosecutions (DPP)
Example 4: and fair investigation not even include the Director of Athletics?  Finally, we believe the
<user>

<assistant>
[1,1,1,1,1]
</assistant>
```

Listing 5: The evaluation model prompt and few-shot examples used to compute the `detection` scores of natural language feature descriptions. `<system></system>`, `<user></user>`, and `<assistant></assistant>` delineate the message roles passed to the evaluation model.

```
<system>
You are an intelligent and meticulous linguistics researcher.

You will be given a certain latent of text formatted as a Semantic Regex. {SEMANTIC_REGEX_DESCRIPTION}

You will then be given several text examples. Your task is to determine which examples possess the latent.

For each example in turn, return 1 if the sentence is correctly labeled or 0 if the tokens are mislabeled.
You must return your response in a valid Python list. Do not return anything else besides a Python list.

</system>

<user>
Semantic Regex explanation: [:field American football position:]

Test examples:
Example 0:<|endoftext|>Getty Images\n\nPatriots tight end Rob Gronkowski had his boss'
Example 1: names of months used in The Lord of the Rings:\n\n"...the
Example 2: Media Day 2015\n\nLSU defensive end Isaiah Washington (94) speaks to the
Example 3: shown, is generally not eligible for ads. For example, videos about recent tragedies,
Example 4: line, with the left side – namely tackle Byron Bell at tackle and guard Amini
</user>

<assitant>
[1,0,1,0,1]
</assistant>

<user>
Semantic Regex explanation: [:symbol you guys:]

Test examples:
Example 0: if you are<< comfortable>> with it. You<< guys>> support me in many other ways already and
Example 1: birth control access<|endoftext|> but I assure you<< women>> in Kentucky aren't laughing as they
 struggle
Example 2:'s gig! I hope you guys<< LOVE>> her, and<< please>> be nice,
Example 3:American, told<< Hannity>> that "you<< guys>> are playing the race card."
Example 4:<< the>><|endoftext|>I want to<< remind>> you all that 10 days ago (director Massimil
</user>

<assistant>
[0,0,0,0,0]
</assistant>

<user>
Semantic Regex explanation: [:symbol of:] [:field Capitalized Word:]

Test examples:
Example 0: climate, Tomblin's Chief of Staff Charlie Lorensen said.\n
Example 1: no wonderworking relics, no true Body and Blood of Christ, no true Baptism
Example 2: Deborah Sathe, Head of Talent Development and Production at Film London,
Example 3: It has been devised by Director of Public Prosecutions (DPP)
Example 4: and fair investigation not even include the Director of Athletics? Finally, we believe the
<user>

<assistant>
[1,1,1,1,1]
</assistant>
```

Listing 6: The evaluation model prompt and few-shot examples used to compute the `detection` scores of semantic regex feature descriptions. `<system></system>`, `<user></user>`, and `<assistant></assistant>` delineate the message roles passed to the evaluation model. At prompt time, {SEMANTIC_REGEX_DESCRIPTION} is replaced with the semantic regex description in Listing 1.

```
<system>
You are an intelligent and meticulous linguistics researcher.

You will be given a certain latent of text, such as "male pronouns" or "text with negative sentiment".

You will be given a few examples of text that contain this latent. Portions of the sentence which strongly
represent this latent are between tokens << and >>.

Some examples might be mislabeled. Your task is to determine if every single token within << and >> is
correctly labeled. Consider that all provided examples could be correct, none of the examples could be
correct, or a mix. An example is only correct if every marked token is representative of the latent

For each example in turn, return 1 if the sentence is correctly labeled or 0 if the tokens are mislabeled.
You must return your response in a valid Python list. Do not return anything else besides a Python list.
```

```
</system>

<user>
Latent explanation: Words related to American football positions, specifically the tight end position.

Test examples:
Example 0:<|endoftext|>Getty Images\n\nPatriots<< tight end>> Rob Gronkowski had his boss'
Example 1: posted<|endoftext|>You should know this<< about>> offensive line coaches: they are large,
demanding<< men>>
Example 2: Media Day 2015\n\nLSU<< defensive>> end Isaiah Washington (94) speaks<< to the>>
Example 3:<< running backs>>," he said. .. Defensive<< end>> Carroll Phillips is improving and his injury
is
Example 4:<< line>>, with the left side – namely<< tackle>> Byron Bell at<< tackle>> and<< guard>> Amini
</user>

<assistant>
[1,0,1,0,1]
</assistant>

<user>
Latent explanation: The word 'guys' in the phrase 'you guys'.

Test examples:
Example 0: if you are<< comfortable>> with it. You<< guys>> support me in many other ways already and
Example 1: birth control access<|endoftext|> but I assure you<< women>> in Kentucky aren't laughing as they
 struggle
Example 2:'s gig! I hope you guys<< LOVE>> her, and<< please>> be nice,
Example 3:American, told<< Hannity>> that "you<< guys>> are playing the race card."
Example 4:<< the>><|endoftext|>I want to<< remind>> you all that 10 days ago (director Massimil
</user>

<assistant>
[0,0,0,0,0]
</assistant>

<user>
Latent explanation: "of" before words that start with a capital letter.

Test examples:
Example 0: climate, Tomblin's Chief Chief<< of>> Staff Charlie Lorensen said.\n
Example 1: no wonderworking relics, no true Body and Blood<< of>> Christ, no true Baptism
Example 2: Deborah Sathe, Head<< of>> Talent Development and Production at Film London,
Example 3: It has been devised by Director<< of>> Public Prosecutions (DPP)
Example 4: and fair investigation not even include the Director<< of>> Athletics? Finally, we believe the
<user>

<assistant>
[1,1,1,1,1]
</assistant>
```

Listing 7: The evaluation model prompt and few-shot examples used to compute the `fuzzing` scores of natural language feature descriptions. `<system></system>`, `<user></user>`, and `<assistant></assistant>` delineate the message roles passed to the evaluation model.

```
<system>
You are an intelligent and meticulous linguistics researcher.

You will be given a certain latent of text formatted as a Semantic Regex. {SEMANTIC_REGEX_DESCRIPTION}

You will be given a few examples of text that contain this latent. Portions of the sentence which strongly
represent this latent are between tokens << and >>.

Some examples might be mislabeled. Your task is to determine if every single token within << and >> is
correctly labeled. Consider that all provided examples could be correct, none of the examples could be
correct, or a mix. An example is only correct if every marked token is representative of the latent

For each example in turn, return 1 if the sentence is correctly labeled or 0 if the tokens are mislabeled.
You must return your response in a valid Python list. Do not return anything else besides a Python list.

</system>

<user>
Semantic Regex explanation: [:field American football position:]

Test examples:
Example 0:<|endoftext|>Getty Images\n\nPatriots<< tight end>> Rob Gronkowski had his boss'
Example 1: posted<|endoftext|>You should know this<< about>> offensive line coaches: they are large,
demanding<< men>>
Example 2: Media Day 2015\n\nLSU<< defensive>> end Isaiah Washington (94) speaks<< to the>>
Example 3:<< running backs>>," he said. .. Defensive<< end>> Carroll Phillips is improving and his injury
is
Example 4:<< line>>, with the left side – namely<< tackle>> Byron Bell at<< tackle>> and<< guard>> Amini
</user>

<assistant>
[1,0,1,0,1]
</assistant>
```

```
<user>
Semantic Regex explanation: [:symbol you guys:]

Test examples:
Example 0: if you are<< comfortable>> with it. You<< guys>> support me in many other ways already and
Example 1: birth control access<|endoftext|> but I assure you<< women>> in Kentucky aren't laughing as they
  struggle
Example 2:'s gig! I hope you guys<< LOVE>> her, and<< please>> be nice,
Example 3:American, told<< Hannity>> that "you<< guys>> are playing the race card."
Example 4:<< the>><|endoftext|>I want to<< remind>> you all that 10 days ago (director Massimil
</user>

<assistant>
[0,0,0,0,0]
</assistant>

<user>
Semantic Regex explanation: [:symbol of:] [:field Capitalized Word:]

Test examples:
Example 0: climate, Tomblin's Chief Chief<< of>> Staff Charlie Lorensen said.\n
Example 1: no wonderworking relics, no true Body and Blood<< of>> Christ, no true Baptism
Example 2: Deborah Sathe, Head<< of>> Talent Development and Production at Film London,
Example 3: It has been devised by Director<< of>> Public Prosecutions (DPP)
Example 4: and fair investigation not even include the Director<< of>> Athletics? Finally, we believe the
<user>

<assistant>
[1,1,1,1,1]
</assistant>
```

Listing 8: The evaluation model prompt and few-shot examples used to compute the `fuzzing` scores of semantic regex feature descriptions. `<system></system>`, `<user></user>`, and `<assistant></assistant>` delineate the message roles passed to the evaluation model. At prompt time, {SEMANTIC_REGEX_DESCRIPTION} is replaced with the semantic regex description in Listing 1.

```
<system>
You are tasked with building a database of sequences that best represent a specific concept.
To create this, you will review a dataset of varying sequences and rate each one according to how much the
concept is expressed.

For each sequence, assign a rating based on this scale:

0: The concept is not expressed.
1: The concept is vaguely or partially expressed.
2: The concept is clearly and unambiguously present.

Use conservative ratings. If uncertain, choose a lower rating to avoid including irrelevant sequences in
your database.
If no sequence expresses the concept, rate all sequences as 0.

Each sequence is identified by a unique ID. Provide your ratings as a Python dictionary with sequence IDs
as keys and their ratings as values.

Example Output: {{"14": 0, "15": 2, "20": 1, "27": 0}}

Output only the dictionary – no additional text, comments, or symbols.
</system>
```

Listing 9: The evaluation model prompt used to compute the `responsiveness` and `purity` scores of natural language feature descriptions. `<system></system>` delineates the message roles passed to the evaluation model.

```
<system>
You are tasked with building a database of sequences that best represent a specific concept.
To create this, you will review a dataset of varying sequences and rate each one according to how much the
concept is expressed.

The concept will be written as a Semantic Regex. {SEMANTIC_REGEX_DESCRIPTION}

For each sequence, assign a rating based on this scale:

0: The concept is not expressed.
1: The concept is vaguely or partially expressed.
2: The concept is clearly and unambiguously present.

Use conservative ratings. If uncertain, choose a lower rating to avoid including irrelevant sequences in
your database.
If no sequence expresses the concept, rate all sequences as 0.

Each sequence is identified by a unique ID. Provide your ratings as a Python dictionary with sequence IDs
as keys and their ratings as values.
```

```
Example Output: {{"14": 0, "15": 2, "20": 1, "27": 0}}

Output only the dictionary – no additional text, comments, or symbols.
</system>
```

Listing 10: The evaluation model prompt used to compute the `responsiveness` and `purity` scores of semantic regex feature descriptions. `<system></system>` delineates the message roles passed to the evaluation model. At prompt time, {SEMANTIC_REGEX_DESCRIPTION} is replaced with the semantic regex description in Listing 1.

```
<system>
You are tasked with building a database of sequences that best represent a specific concept.
To create this, you will generate sequences that vary in style, tone, context, length, and structure, while
 maintaining a clear connection to the concept.
The concept does not need to be explicitly stated in each sequence, but each should relate meaningfully to
it. Be creative and explore different ways to express the concept.

Here are examples of how different concepts might be expressed:

Concept: "German language" – Sequences might include German phrases, or sentences.
Concept: "Start of a Java Function" – Sequences might include Java code snippets defining a function.
Concept: "Irony" – Sequences might include ironic statements or expressions.

Provide your sequences as strings in a Python List format.

Example: ["This is a first example sequence.", "Second example sequence but it is much longer also there
are somy typos in it. wjo told you that I can type?"]

Output only the Python List object, without any additional comments, symbols, or extraneous content.
</system>
```

Listing 11: The evaluation model prompt used to compute the `clarity` scores of natural language feature descriptions. `<system></system>` delineates the message roles passed to the evaluation model.

```
<system>
You are tasked with building a database of sequences that best represent a specific concept.
To create this, you will generate sequences that vary in style, tone, context, length, and structure, while
 maintaining a clear connection to the concept.

The concept will be expressed as a Semantic Regex. {SEMANTIC_REGEX_DESCRIPTION}
Be creative and explore different ways to express the concept, while faithfully expressing the semantic
regex.

Here are examples of how different concepts might be expressed:

Concept: "[:topic German Language:]" – Sequences might include German phrases, or sentences.
Concept: "@{{Java}}(functions)" – Sequences might include Java code snippets defining a function.
Concept: "[:lexeme irony:]" – Sequences that include the string 'irony', 'ironic', 'ironically', etc.

Provide your sequences as strings in a Python List format.

Example: ["This is a first example sequence.", "Second example sequence but it is much longer also there
are somy typos in it. wjo told you that I can type?"]

Output only the Python List object, without any additional comments, symbols, or extraneous content.
</system>
```

Listing 12: The evaluation model prompt used to compute the `clarity` scores of semantic regex feature descriptions. `<system></system>` delineates the message roles passed to the evaluation model. At prompt time, {SEMANTIC_REGEX_DESCRIPTION} is replaced with the semantic regex description in Listing 1.

