# OpenReview forum: "Semantic Regexes: Auto-Interpreting LLM Features with a Structured Language"
_ICLR.cc/2026/Conference — ICLR 2026 Poster_

### Official Review · Reviewer_riWE · 2025-10-27

**Soundness:** 2
**Presentation:** 3
**Contribution:** 2
**Rating:** 6
**Confidence:** 2

**Summary:**

The authors propose semantic-regex, a structured language framework, aimed to improve expressiveness and clarity of feature description protocols. This aims to address the typically verbose natural language output of feature description methods. The authors use the Neuropedia suite for extracting feature descriptions and the FADE framework for evaluation and find that semantic-regex perform broadly as well as natural language feature description methods, while sometimes being of higher clarity.

**Strengths:**

- Using structured language for re-framing the problem of feature description is an interesting idea, especially for improving clarity (as also supported by some of the evaluations).
- The methods are clearly defined and appear to be reproducible with reasonable efforts.
- The human user study provides a complementary contribution, highlighting differences in scoring between the max-acts baseline and semantic-regex.

**Weaknesses:**

- The framework appears to work with binary assessments of feature presence, i.e. the semantic regex indicated presence/absence of a feature description. But, simple pattern matching is insufficient to capture some of the diverse patterns learned by today’s foundation models, e.g. nonregular language.
- A clear presentation and discussion of limitations is missing. Specifically, the quantitative experiments (Fig. 2 and Fig. 5 of the human user study) remained vague regarding the differences and benefits/disadvantages that semantic-regex would bring compared to previous natural language-based approaches. What are the limits of semantic regex, i.e., in the context of polysemantic feature descriptions, or features encoding pragmatics, see [Kop25].
- The paper does not effectively address how the necessary semantic regex language should be developed, how to agree on primitives and grammatical structure?
- Limited model ablations are included, e.g. what would be the effect of another explainer model (so far only gpt-4o-mini is included) or another evaluator model (also gpt-4o-mini is used). This may be a poor experimental choice as the explainer and evaluation are the same (closed source) model.
- Overall, the contribution is limited in scope to reformulating existing methods to using a specific language format (semantic regex) for defining and communicating feature descriptions. The paper does not provide a novel method for detecting feature descriptions and reuses the methods defined and implemented via Neuronpedia [Lin, 2023; Bills et al. (2023)]. The proposed method should thus be seen as an extension to these methods.

**Refs:**

[Kop25] Kopf, L., Feldhus, N., Bykov, K., Bommer, P. L., Hedström, A., Höhne, M. M. C., & Eberle, O. (2025). Capturing Polysemanticity with PRISM: A Multi-Concept Feature Description Framework. arXiv preprint arXiv:2506.15538.

**Questions:**

**Questions**
- How much of a challenge is the explainer’s ability to parse regex correctly?
- The used evaluation framework FADE (Puri et al., 2025), relies on generating synthetic control samples, e.g. for the faithfulness metric.
- How well can models generate synthetic data from semantic regex feature description compared to natural language ones? Here a comparative investigation would allow revealing the advantages and limits of the proposed framework.


**Typos**
- “regexes as as” (l. 61) -> “are as”
- “has found structure in the concepts features represent.” (l.90)
- “that semantic regexes make activation patterns explicit and express them via example.” (l. 424)

---

> ### Author Response · Authors · 2025-11-25
> **Response to riWE**
>
> Thank you for this thorough review! Your suggestions strengthened the paper, and we added an ablation study and dedicated limitations section — overall changes are in the general comment above and specific responses are inline below.
>
> ---
> > The framework appears to work with binary assessments of feature presence … simple pattern matching is insufficient to capture some of the diverse patterns learned by today’s foundation models, e.g. nonregular language.
>
> While we analogize semantic regexes to regular expressions to give readers intuition for their pattern-based descriptions, the semantic regex language is not a regular language. It includes abstract primitives that can express non-regular patterns, such as balanced parentheses:`[:field balanced parentheses:]`. It can also describe contextual or long-range dependencies using the `context` and `concatenation` modifiers. **We clarify that the semantic regex language is not a regular language in Section 3.**
>
> While standard in the field, the interpretability metrics we adopt do rely on binary judgements. We agree this is an oversimplification and **we discuss more nuanced evaluation metrics in Section 6**.
>
> ---
> > …discussion of limitations is missing...
>
> Thank you for pointing this out! **We added a dedicated limitation section (Section 6)**.
>
> ---
> > The paper does not effectively address how the necessary semantic regex language should be developed…
>
> We developed the semantic regex language using a grounded-theory approach (Corbin & Strauss, 1998) where we manually surveyed thousands of LLM features and identified recurring activation patterns. We defined a language component when it captured a recurring pattern, enabled us to describe features that could not be expressed previously, and preserved the intelligibility of the language. We added components to the language until it could describe all the features we analyzed. **We added our language development process to Section 3.**
>
> ---
> > Limited model ablations are included, e.g. what would be the effect of another explainer model...
>
> Thank you for this suggestion. To validate our results with a more capable model, we run a new ablation study using GPT-4o. We find that the relative performance between feature description methods stays the same regardless whether we use GPT-4o-mini or GPT-4o. These results suggest that GPT-4o-mini can produce high-quality feature descriptions at only 6% the cost of GPT-4o.
>
> However, using the same model to generate and evaluate descriptions is standard in automated interpretability (e.g., Bills et al. 2023; Gur-Arieh et al. 2025), and prior work (Chen et al. 2025) shows that perceived self-preference often results from superior performance rather than bias. Moreover, since we use the same explainer/evaluator setup for both semantic regexes and natural language, any potential self-preference bias would affect both and not change the relative comparison that is central to our non-inferiority claim.
>
> **The new ablation and a discussion of our model choices are in Section 4 and Appendix C.**
>
> ---
> > The paper reuses the methods defined and implemented via Neuronpedia...
>
> Yes, we intentionally design semantic regexes to slot into existing automated interpretability pipelines. This ensures that as pipelines improve via new prompting methods, sampling techniques, or additions (like PRISM from Kopf et al., 2025), so will semantic regexes. **We clarify this design choice in Section 4 and Section 6.**
>
> ---
> > How much of a challenge is the explainer’s ability to parse regex correctly?... How well can models generate synthetic data from semantic regex feature description...
>
> These are great questions. We do observe "grammatical" errors in both the explainer (e.g., applying the wrong primitive) and evaluator (e.g., misinterpreting the meaning of a primitive and generating a non-activating example). However, semantic regexes perform on par with natural language and achieve the strongest results on `clarity` (i.e., the metric most dependent on synthetic generation). This suggests that despite occasional grammatical errors, models can learn the semantic regex language. **We discuss limitations of having to learn a novel language in Section 6 and release our semantic regexes and metric computations for the camera ready.**

---

### Official Review · Reviewer_CRyN · 2025-10-30

**Soundness:** 4
**Presentation:** 3
**Contribution:** 3
**Rating:** 8
**Confidence:** 3

**Summary:**

This paper proposes to explain features with semantic regular expresssions, which can provide more consistant,ni concise, and precise explanations compared to ambigious natural language explanations. I think this is a solid work and provides a practical tool for managing and standardizing feature descriptions. While I find the core contribution strong, there are some limitations in validation and minor issues to address.

**Strengths:**

- The paper is well-structured and clearly written, which is very easy to follow. The methods used in this paper are straightforward as well.

- The categorization of regex components is clean and comprehensive. The progression from [:symbol:] (exact matches) to [:lexeme:] (syntactic variants) to [:field:] (semantic variants), combined with context modifiers, provides appropriate levels of abstraction for describing features.

- The experimental validation is thorough, incorporating multiple automated metrics (detection, fuzzing, clarity, responsiveness, purity, faithfulness) across different models (GPT-2, Gemma-2) and feature sets. The results effectively demonstrate improved precision and consistency at both syntactic and semantic levels.

**Weaknesses:**

- Human sanity check only involves 12 features. This raises questions about the method's reliability across all features.
- Lacks discussion of failur models where semantic regexes might fail.

**Questions:**

- L61, typo "as as accurate as"
- Is the example for quantifiers in Figure 1 correct? Should "it is a da isy" be "it is a [color] daisy"
- L242, missing "measures" in the sentence?
- How were the 12 "accurate" features selected for the user study? What percentage of automatically generated descriptions were deemed accurate? This information is crucial for assessing the method's practical reliability.

---

> ### Author Response · Authors · 2025-11-25
> **Response to Reviewer CRyN**
>
> Thank you for your review! Your comments helped us clarify important details and inspired us to add a dedicated limitations section – overall changes are in the general comment above and specific responses are inline below.
>
> ---
> > Lacks discussion of failure models where semantic regexes might fail.
>
> Thank you for pointing this out! **We added a dedicated limitation section (Section 6)**. Key limitations include (see Section 6 for full details):
>
> * While semantic regexes can describe simple polysemanticity using the OR operator, they often produce incoherent descriptions for higher degrees of polysemanticity. However, since they are agnostic to the generation pipeline, they could easily slot into new techniques that disentangle polysemantic activations (Kopf et al., 2025).
>
> * While we designed semantic regexes to be concise, they can become overly brief. Improved prompts, additional language primitives, or validation loops (Shaham et al., 2024) could better balance between conciseness and expressivity.
>
> * Multiple valid semantic regexes can describe the same feature. As in programming languages, this is not inherently harmful, but style heuristics could improve readability.
>
> * To keep the syntax simple, we leave some components underspecified, but this can cause models to make inconsistent assumptions between features. Future language components could reduce this ambiguity.
>
> * Semantic regexes require models to learn the semantic regex language from only a brief specification and few examples. As a result, we observe "grammatical" errors (e.g., applying the wrong primitive). Nevertheless, semantic regexes perform similarly to natural language, suggesting models are able to sufficiently learn the language.
>
> ---
> > Human sanity check only involves 12 features. This raises questions about the method's reliability across all features.
>
> > How were the 12 "accurate" features selected for the user study? What percentage of automatically generated descriptions were deemed accurate? This information is crucial for assessing the method's practical reliability.
>
> Since we evaluate the reliability of semantic regexes at scale in Section 5.1, the user study is not meant to statistically measure reliability. Instead, the study provides qualitative insight into how different feature description formats impact experts’ interpretations. By using 12 diverse features, we avoid overburdening our valuable experts, and, importantly, reveal interesting insights.
>
> We aim to select features with both accurate semantic regex and natural language descriptions, so we first filtered to features with a `detection` score greater than 75%. From these, we manually selected 12 diverse features across layers (5 early, 4 middle, 3 late) and activation types: 2 symbol, 1 lexeme, 3 field, 2 context-dependent, and 4 complex compositional patterns. We did not aim to identify all accurate features, but a representative, high-quality, diverse set for studying human interpretation.
>
> We also agree that a larger crowdsourced study on human interpretation of feature descriptions could provide interesting insight into the value of feature descriptions in human interpretation pipelines and how LLM-as-a-judge evaluations compare to human preferences in this domain.
>
> **We better motivate and describe our study design in Section 5.4 and Appendix E. We discuss additional studies in Section 6. We will release the user study data and features for the camera-ready.**
>
> ---
> > Is the example for quantifiers in Figure 1 correct? Should "it is a da isy" be "it is a [color] daisy"
>
> Yes, in `[:symbol a:] [:field color:]? [:field flower:]`, the `?` makes the color optional, so the semantic regex matches both `a red rose` and `a da isy`.

---

### Official Review · Reviewer_YyVu · 2025-11-01

**Soundness:** 3
**Presentation:** 3
**Contribution:** 2
**Rating:** 4
**Confidence:** 2

**Summary:**

This paper introduces semantic regexes, a structured language for describing LLM features. The authors demonstrate through experiments that semantic regexes perform on par with natural language feature descriptions while being more concise, consistent, and able to reflect feature complexity. A user study shows that semantic regexes benefit interpreters’ ability to understand and reason about LLM features.

**Strengths:**

This paper studies an interesting task of generating language descriptions for LLM features. The paper provides concrete examples to illustrate the methodology and experimental results. The implementation and experimental settings have been presented in detail. The insights presented in Section 5.3 sound intriguing.

**Weaknesses:**

There seems to be a lack of interesting insights apart from Section 5.3. The facts that regexes are more concise and consistent than natural languages seem quite obvious given that regexes are a more restricted form of languages.

The experiments can be enhanced by evaluating more datasets and LLMs. It is unclear why the evaluator model, gpt-4o-mini, is the same as the explainer model since this may introduce bias. Would it be better to switch to a larger model like gpt-4o?

The population size of the user study, 24, seems too small for drawing conclusions.

**Questions:**

What is the rationale for using GPT-4o-mini as the evaluator model?

How many repeated runs have been conducted in the experiments?

---

> ### Author Response · Authors · 2025-11-25
> **Response to Reviewer YyVu**
>
> Thank you for your thoughts! Your questions have helped us strengthen our writing and inspired an additional ablation study – overall changes are in the general comment above and specific details are inline below.
>
> ---
> > The fact that regexes are more concise and consistent than natural languages seem quite obvious given that regexes are a more restricted form of languages.
>
> Yes, we explicitly designed semantic regexes to be concise and consistent. But, the non-obvious question was whether we could add this structure and still express the full range of LLM features. Our results show that semantic regexes perform on par with natural language feature descriptions, meaning we gain the benefits of structure (from consistency to new analyses of feature complexity) without sacrificing descriptive power. Conciseness also supports human interpretation. While natural language descriptions often included distracting details, semantic regexes’ conciseness made the pattern more obvious.
>
> **We emphasize these points in Section 5.2, Section 5.4, and Section 6.**
>
> ---
> > The experiments can be enhanced by evaluating more datasets and LLMs. It is unclear why the evaluator model, gpt-4o-mini, is the same as the explainer model since this may introduce bias. Would it be better to switch to a larger model like gpt-4o?
>
> > What is the rationale for using GPT-4o-mini as the evaluator model?
>
> This is an excellent question.
>
> Using the same model to generate and evaluate descriptions is standard in automated interpretability (e.g., Bills et al. 2023; Gur-Arieh et al. 2025). Verification is easier than generation, making a model a reliable critic of its own outputs, and the model calls are independent, so the evaluator has no memory of generating the description. Prior work (Chen et al. 2025) also shows that perceived self-preference often results from superior performance rather than bias. Moreover, since we use the same explainer/evaluator setup for both semantic regexes and natural language, any potential self-preference bias would affect both and not change the relative comparison that is central to our non-inferiority claim.
>
> To validate our results with a more capable model, we run a new ablation study using GPT-4o. We find that the relative performance between feature description methods stays the same regardless whether we use GPT-4o-mini or GPT-4o. These results suggest that GPT-4o-mini can produce high-quality feature descriptions at only 6% the cost of GPT-4o.
>
> **The new ablation and a discussion of our model choices are in Section 4 and Appendix C.**
>
> ---
> > The population size of the user study, 24, seems too small for drawing conclusions.
>
> Our goal was to understand how semantic regexes impact real-world practitioners, so we sampled 24 AI experts who represent people that would use semantic regexes. While 24 participants would be low if we were trying to make statistical claims about the accuracy of semantic regexes, the user study is designed to provide qualitative insight into semantic regexes’ impact on real-world experts. It is consistent with comparable human-computer interaction studies (Caine, 2016), and, importantly, was sufficient to draw insights about the benefits and limitations of semantic regexes.
>
> We also agree that a larger crowdsourced study on human interpretation of feature descriptions could provide interesting insight into the value of feature descriptions in human interpretation pipelines and how LLM-as-a-judge evaluations compare to human preferences in this domain.
>
> **We discuss these points in Section 5.4, Section 6, and Appendix E**.
>
> ---
> > How many repeated runs have been conducted in the experiments?
>
> Rather than repeating description generation on the same feature, we intentionally focus our resources on generating descriptions for a diverse set of LLM features. In total we evaluate feature descriptions on over 6000 features and generate descriptions for an additional 25,000. However, we agree that additional repetitions could strengthen our understanding of feature description pipelines.
>
> **We discuss this in Section 6**.

---

### Author Response · Authors · 2025-11-25
**Response to All Reviewers**

Thank you for your insightful (and human-written) reviews!

We are encouraged that the reviewers think *“re-framing the problem of feature description is an interesting idea”* (riWE), our use of structured language to measure feature complexity is *“intriguing”* (YyVu), the semantic regex language is *“clean and comprehensive”* (CRyN), and *“the experimental validation is thorough”* (CRyN).

Based on your suggestions, we have strengthened the paper significantly. We have added a new ablation study, dedicated limitations section, and expanded discussions on our methodology. We summarize the major updates here (changes highlighted in the PDF) and respond individually to each reviewer as comments below. We believe these revisions address the reviewers' concerns and substantially improve the paper.


**Model Ablation Study (Appendix C)**

Reviewers YyVu and riWE raised important questions about the choice of GPT-4o-mini as the explainer and evaluator model. To address concerns regarding model capability, we conducted a new ablation study on a subset of GPT-2 features using GPT-4o. We find that the relative performance between feature description methods remains the same regardless of whether we use GPT-4o-mini or GPT-4o. This reinforces our claim that semantic regexes perform on par with natural language while offering additional benefits. It also suggests that GPT-4o-mini produces high-quality descriptions at approximately 6% of the cost of GPT-4o, lowering the barrier for large-scale semantic regex generation.


**New Limitations Section (Section 6)**

Based on feedback from CRyN and riWE, we have added a comprehensive discussion of the limitations of semantic regexes. In particular, we describe the inherent tradeoffs in the design of the semantic regex language (e.g., language simplicity vs. specificity), semantic regexes’ ability to handle simple (but not high degrees of) polysemanticity, and expansions to our evaluations. This section helps readers understand and appropriately apply semantic regexes, and it also lays out exciting future work on structured languages for interpretability, pluralistic evaluations in automated interpretability pipelines, and user studies that reveal the similarity between human and model preferences in this domain.


**Clarifications on Methodology and Design**

* *Semantic Regex Language Development (Section 3)*: In response to riWE, we clarified that we developed the semantic regex language using a grounded-theory methodology, iteratively deriving components from empirical analysis of thousands of real LLM features.

* *User Study Goals and Process (Section 5.4 and Appendix E)*: Addressing YyVu and CRyN, we justified our user study design. Since it was designed to provide qualitative insight into semantic regexes’ impact on real-world experts (not statistical measures of reliability), our sample size (24 AI experts and 12 LLM features) follows standard HCI practices and, importantly, is sufficient to draw insights about the benefits and limitations of semantic regexes. We also describe our feature selection process and how we focused on identifying a diverse set of features with accurate natural language and semantic regex description.


**GitHub Repository and Interactive Interface**

To increase the transparency of our method, we will include our GitHub repository and interactive web interface in the camera-ready version of the paper. These resources allow the community to explore our results themselves. They contain the complete data from our experiments, including the generated semantic regex and natural language descriptions for every feature we analyzed (over 25,000 total features). They also provide the evaluation scores and the intermediate generation data and model judgements used to compute them.

---

### Meta-Review · Area_Chair_jZnv · 2025-12-28

**Summary:**

This paper introduces Semantic Regexes, a structured language framework designed to produce precise, concise, and consistent descriptions of LLM features by combining semantic primitives with contextual modifiers. The reviewers praised the work's clean language categorization and its novel approach to re-framing feature description, noting that the method effectively balances structure with expressivity. While initial concerns were raised regarding the reliance on GPT-4o-mini for both generation and evaluation and the limited scale of the human user study, the authors successfully addressed these during the rebuttal by performing an ablation study with GPT-4o that confirmed the stability of their results. Furthermore, the authors expanded the manuscript to include a comprehensive discussion on the limitations of the framework regarding polysemanticity and non-regular patterns. Given the thorough experimental validation and the authors' effective response to feedback regarding model capability and study design, I recommend accepting this paper for ICLR 2026.

**Reviewer Concerns:**

Concerns addressed by the rebuttal:
- Model Ablation Study
- New Limitations Section
- Clarifications on Methodology and Design

**Reviewer Scores:**

I believe most reviewers would maintain their score if they had been able to participate fully in the discussion.

---

### Decision · Program_Chairs · 2026-01-26

Accept (Poster)